# Inhibition of interleukin-1β reduces myelofibrosis and osteosclerosis in mice with *JAK2*-V617F driven myeloproliferative neoplasm

Shivam Rai[1], Elodie Grockowiak[2,3,4], Nils Hansen[1], Damien Luque Paz[1], Cedric B. Stoll[1], Hui Hao-Shen[1], Gabriele Mild-Schneider[1], Stefan Dirnhofer[5], Christopher J. Farady [6], Simón Méndez-Ferrer[2,3,4] & Radek C. Skoda [1] ✉

Interleukin-1β (IL-1β) is a master regulator of inflammation. Increased activity of IL-1β has been implicated in various pathological conditions including myeloproliferative neoplasms (MPNs). Here we show that IL-1β serum levels and expression of IL-1 receptors on hematopoietic progenitors and stem cells correlate with *JAK2*-V617F mutant allele fraction in peripheral blood of patients with MPN. We show that the source of IL-1β overproduction in a mouse model of MPN are *JAK2*-V617F expressing hematopoietic cells. Knockout of *IL-1β* in hematopoietic cells of *JAK2*-V617F mice reduces inflammatory cytokines, prevents damage to nestin-positive niche cells and reduces megakaryopoiesis, resulting in decrease of myelofibrosis and osteosclerosis. Inhibition of IL-1β in *JAK2*-V617F mutant mice by anti-IL-1β antibody also reduces myelofibrosis and osteosclerosis and shows additive effects with ruxolitinib. These results suggest that inhibition of IL-1β with anti-IL-1β antibody alone or in combination with ruxolitinib could have beneficial effects on the clinical course in patients with myelofibrosis.

Myeloproliferative neoplasms (MPNs) are clonal disorders of the hematopoietic stem cell (HSC), caused by somatic mutations in *JAK2*, *MPL*, or *CALR* resulting in increased proliferation of the erythroid, megakaryocytic, and myeloid lineages[1,2]. MPN can manifest in one of three phenotypic subtypes, polycythemia vera (PV), essential thrombocythemia (ET), and primary myelofibrosis (PMF)[1]. Myelofibrosis is characterized by increased deposition of reticulin and/or collagen fibers, megakaryocytic hyperplasia with atypical features and extramedullary hematopoiesis in spleen and liver[3]. Abnormal megakaryocytes play a key role in the pathogenesis of myelofibrosis by releasing profibrotic factors such as TGF-β and producing excessive amounts of fibronectin, laminin, and collagen[3].

Inflammatory cytokines are known to be elevated in MPN patients, with highest levels present in patients with advanced stages of PMF[4,5].

Studies in patients and preclinical mouse models of MPN have linked inflammation with MPN progression to myelofibrosis[6–8]. IL-1β is considered a master regulator of inflammation that controls the production of multiple pro-inflammatory cytokines and induce its own expression via a positive feedback loop in an autocrine or paracrine manner[9,10]. *JAK2*-V617F increases IL-1β levels in a mouse model of MPN and IL-1β was implicated in damaging Schwann cells and sympathetic nerve fibers that are required for maintaining nestin-positive stromal cells in the bone marrow HSC niche[11]. Moreover, chronic exposure to IL-1β favors HSC differentiation towards myeloid lineages at the expense of self-renewal[12].

IL-1β and IL-1α, another member of the IL-1 cytokine family, both signal through the same receptor heterodimer by binding to IL-1R1, which induces a conformational change favoring the recruitment of

[1]Department of Biomedicine, Experimental Hematology, University Hospital Basel, University of Basel, 4031 Basel, Switzerland. [2]Wellcome-MRC Cambridge Stem Cell Institute, Cambridge CB2 0AW, UK. [3]Department of Hematology, University of Cambridge, Cambridge CB2 0AW, UK. [4]National Health Service Blood and Transplant, Cambridge Biomedical Campus, Cambridge CB2 0AW, UK. [5]Department of Pathology, University Hospital Basel, 4031 Basel, Switzerland. [6]Novartis Institutes for BioMedical Research Forum 1, Basel, Switzerland. ✉e-mail: radek.skoda@unibas.ch

co-receptor, IL-1RAcP[10]. The resulting trimeric complex via conserved cytosolic Toll- and IL-1R-like (TIR) domain rapidly assembles two intracellular signaling proteins, myeloid differentiation primary response gene 88 (MYD88) and interleukin-1 receptor-activated protein kinase (IRAK) 4 and initiating intracellular signaling[13]. IL-1 signaling in the extracellular space is regulated by diverse mechanisms at multiple levels including receptor antagonists, and soluble or plasma membrane-anchored receptors or co-receptors, reflecting the need for tight regulation of the IL-1 system[14,15]. In particular, the IL-1 receptor antagonist (IL-1RA) binds IL-1R1 with higher affinity than IL-1β or IL-1α and limits the recruitment of IL-1RAcP, thereby reducing IL-1 inflammatory signaling[10].

The discovery of the *JAK2*-V617F mutation in MPN has led to the development and approval of a JAK1/2 inhibitor, ruxolitinib for the treatment of PMF patients with splenomegaly and constitutional symptoms[16]. While ruxolitinib reduces the production of circulating pro-inflammatory cytokines, it has shown little effect on myelofibrosis progression[17]. Recently, therapies targeting inflammatory pathways beyond JAK1/2 inhibition has shown promise in reducing myelofibrosis[7,8].

In this study, we examine the functional relevance of IL-1β in MPN pathogenesis using genetic and pharmacological approaches. Our study reveals that *JAK2*-V617F mutation in MPN patients correlates with elevated production of IL-1β. Genetic deletion of *IL-1β* from *JAK2*-V617F mutant cells, or pharmacological inhibition of IL-1β are effective in reducing myelofibrosis and osteosclerosis in a preclinical mouse model of MPN. Similar results have been obtained in a study by Dr. Mohi and colleagues[18].

## Results

### *JAK2*-V617F is associated with increased expression of IL-1 in MPN patients

To assess the status of IL-1 signaling in MPN patients, we measured IL-1β and IL-1 receptor antagonist (IL-1RA) expression in a cohort of 120 MPN patients with *JAK2*-V617F mutation and 20 normal controls (NC) (Fig. 1a, b and Supplementary Data 1). Overall, serum IL-1β and IL-1RA levels were elevated in MPN patients compared to NC (Fig. 1a, b) and within the MPN group, PV and PMF patients showed higher serum levels of IL-1β and IL-1RA than ET patients (Fig. 1a, b). IL-1β and IL-1RA levels correlated with *JAK2*-V617F allele burden in DNA from peripheral blood granulocytes, whereas no correlation with allele burden was found for other pro-inflammatory cytokines (Supplementary Fig. 1a, b). *IL1B* (*IL-1β*) mRNA and *IL1RN* (*IL-1RA*) mRNA expression in granulocytes was higher in MPN patients than NC (Fig. 1a, b, lower panel) and PV patients showed the highest *IL1B* mRNA within the MPN group (Fig. 1a, b, lower panel). *IL1B* mRNA expression also correlated with *JAK2*-V617F allele burden in granulocytes, but no correlation of *IL1RN* expression with *JAK2*-V617F allele burden was found. IL-1β is synthesized as an inactive pro-protein, pro-IL-1β, which is cleaved and activated intracellularly by inflammasome mediated caspase-1 activity[14]. We found *caspase1* mRNA expression to be elevated in granulocytes of PV patients compared to NC, but *caspase1* mRNA expression in MPN patients did not correlate with *JAK2*-V617F allele burden (Supplementary Fig. 1c). Serum TGF-β1 levels were elevated in MPN patients (Supplementary Fig. 3a), and showed a weak negative correlation with *JAK2*-V617F allele burden (Supplementary Fig. 3a), but no correlation between TGF-β1 and IL-1β serum levels was found (Supplementary Fig. 3b).

IL-1 signaling requires the formation of a complex between the ligands (IL-1β or IL-1α) and the interleukin-1 receptor, consisting of a dimer between IL-1R1 and interleukin-1 receptor accessory protein (IL-1RAcP)[14]. We examined the proportion of IL-1R1 and IL-1RAcP-positive hematopoietic stem cells (HSCs) and progenitors (HSPCs) in peripheral blood of MPN patients by flow cytometry. The gating strategy and the overall frequencies of HSPCs in peripheral blood are shown in Supplementary Fig. 2a, b and the cutoff for IL-1R1 and IL-1RAcP

positivity is shown in Supplementary Fig. 2c. We found approximately a 3-fold increase in the frequency of IL-1R1[+] and IL-1RAcP[+] HSCs and HSPCs in MPN patients compared to NC (Fig. 1c, d). We also found a strong correlation between *JAK2*-V617F allele burden in granulocytes and the percentages of IL-1R1[+] or IL-1RAcP[+] HSCs and HSPCs in peripheral blood (Fig. 1c, d, and Supplementary Fig. 2d, e), suggesting that the expression of *JAK2*-V617F may trigger the expansion of IL-1R1[+] or IL-1RAcP[+] HSPCs in MPN patients. To further address the relevance of IL-1 signaling in MPN pathogenesis, we analyzed previously published gene expression microarray dataset of peripheral blood CD34[+] HSPCs from *JAK2*-V617F[+] PMF patients and bone marrow CD34[+] HSPCs from normal controls[19]. Gene set enrichment analysis (GSEA) revealed significant enrichment for IL-1R pathway (Fig. 2a) in PMF patients with higher expression of IL-1R pathway target genes compared to normal controls (Fig. 2b). Overall, these results show a good correlation between *JAK2*-V617F and increased IL-1 signaling in MPN patients.

We have extensively characterized an inducible *SclCre^ER;JAK2*-V617F (*VF*) mouse model that faithfully captures many aspects of MPN including progression to myelofibrosis[20,21]. In *VF* mice we also observed upregulated expression of IL-1R pathway genes by RNA sequencing of sorted HSCs, bipotent megakaryocyte-erythroid precursors (pre-MegE) and megakaryocyte progenitors (MkP) (Fig. 2c). Thus, IL-1 signaling is upregulated in *JAK2*-V617F positive MPN patients and in mice expressing *JAK2*-V617F.

### Complete knockout of *IL-1β* in a *JAK2*-V617F MPN mouse model reduces inflammatory cytokines but does not affect the overall course of MPN disease

To further examine the role of IL-1β in MPN pathogenesis, we crossed our *SclCre^ER;JAK2*-V617F (*VF*)[20,21] mice with *IL-1β^-/-* mice[22] and analyzed the resulting double mutant *VF;IL-1β^-/-* mice after induction with tamoxifen. *VF;IL-1β^-/-* mice did not show altered survival, body weight, spleen weight, or grade of reticulin fibrosis versus *VF* single mutant mice, showed only slightly reduced red cell parameters, and overall no significant changes in platelet and leukocyte counts compared to *VF* (Fig. 3a and Supplementary Fig. 4a). Also no differences between *VF* and *VF;IL-1β^-/-* mice were observed in the frequencies of HSCs and HSPCs in bone marrow and spleen (Supplementary Fig. 4b), or in bone marrow, spleen, and liver histology (Supplementary Fig. 5a, b). IL-1β levels in bone marrow lavage and plasma were significantly elevated in *VF* mice compared with wildtype (*WT*) mice, while IL-1β as expected was absent in *VF;IL-1β^-/-* mice (Fig. 3b). IL-1α, the other family member that signals through the same receptor, was not detectable in plasma, but was elevated in bone marrow of *VF* mice along with IL-1β (Fig. 3b). Interestingly, *VF;IL-1β^-/-* mice displayed lower levels of IL-1α than *VF* mice, contrary to the expectation that IL-1α would be upregulated to compensate for the loss of *IL-1β*, but consistent with IL-1α being expressed downstream of IL-1R1 signaling[23]. No differences were found in IL-1Ra levels in bone marrow and plasma between *VF* and *VF;IL-1β^-/-* mice (Supplementary Fig. 4c). While the levels of some pro-inflammatory cytokines were elevated in bone marrow or plasma of *VF* mice compared to *WT*, loss of *IL-1β* resulted in partial or complete normalization of these differences in the bone marrow (Fig. 3c). Taken together, these results show that complete loss of *IL-1β* in this MPN mouse model reduced inflammatory cytokines in the bone marrow, but the expansion of MPN cells and the overall course of disease remained unaffected.

### Loss of *IL-1β* in *JAK2*-V617F mutant hematopoietic cells reduces MPN symptoms and myelofibrosis

Since *VF;IL-1β^-/-* mice lack *IL-1β* expression in all tissues, we examined the effects of *IL-1β* deficiency confined only to hematopoietic cells by performing transplantations of bone marrow cells into lethally irradiated recipient mice (Fig. 4). We found that in contrast to the non-transplanted mice (Fig. 3), platelet and leukocyte counts were lower,

whereas red cell parameters were higher in recipient mice    transplanted with bone marrow from *VF;IL-1β^{−/−}* donors compared to

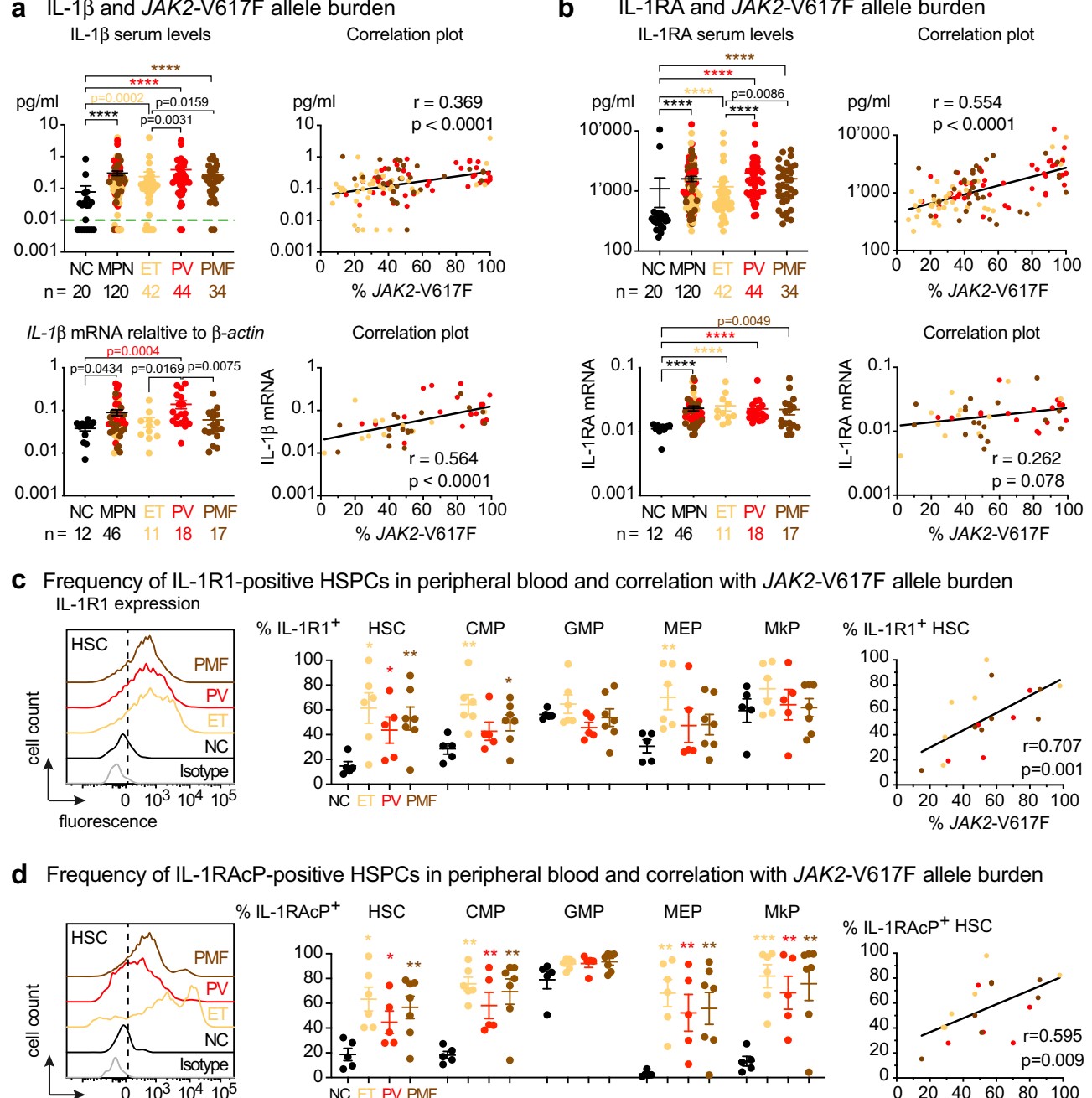

**Fig. 1 | *JAK2*-V617F correlated with increased IL-1 expression in MPN patients.**
**a** Upper panel: Serum IL-1β (pg/ml) in normal controls (NC; *n* = 20) and MPN patients (*n* = 120); ET (*n* = 42), PV (*n* = 44), PMF (*n* = 34). Correlation (*r*) between %*JAK2*-V617F in granulocytes and log transformed serum IL-1β in MPN patients. Limit of detection is shown by dashed green line at y = 0.01 pg/ml. Lower panel: *IL-1β* mRNA expression relative to β-actin in granulocytes of NC (*n* = 12) and MPN patients (*n* = 46); ET (*n* = 11), PV (*n* = 18), PMF (*n* = 17). Correlation between log transformed *IL1B* mRNA expression and %*JAK2*-V617F. **b** Upper panel: Serum IL-1RA (pg/ml) in NC (*n* = 20) and MPN patients (*n* = 120); ET (*n* = 42), PV (*n* = 44), PMF (*n* = 34). Correlation (*r*) between %*JAK2*-V617F and log transformed serum IL-1RA. Lower panel: *IL1RN (IL-1RA)* mRNA expression relative to β-actin in NC and MPN patients. Correlation between log transformed *IL1RN* mRNA expression and %*JAK2*-V617F. Two-tailed unpaired non-parametric Mann−Whitney *t*-test was performed in **a** and **b**. **c** Representative histogram showing the expression of interleukin 1 receptor type 1 (IL-1R1) in peripheral

blood hematopoietic stem cells (HSCs) from isotype control, NC (*n* = 5), ET (*n* = 6), PV (*n* = 5), and PMF (*n* = 7). Bar graph showing the percentages of IL-1R1+ HSC), common myeloid progenitors (CMP), granulocyte macrophage progenitor (GMP), megakaryocyte erythroid progenitor (MEP) and megakaryocyte progenitor (MkP). Graph showing correlation (*r*) between %*JAK2*-V617F and percentages of IL-1R1 + HSCs. **d** Representative histogram showing the expression of interleukin 1 receptor accessory protein (IL-1RAcP) in peripheral blood HSC from NC and MPN patients. Bar graph showing the percentages of IL1RAcP+ HSC, CMP, GMP, MEP, and MkP in NC (*n* = 5), ET (*n* = 6), PV (*n* = 5), and PMF (*n* = 7). Correlation (*r*) between %*JAK2*-V617F and percentages of IL-1RAcP+ HSPCs. Two-tailed unpaired t-test was performed for statistical comparisons in **c** and **d**. Spearman correlation (*r*) and two-tailed *t*-test was performed for correlation analysis in **a**−**d**. All data are presented as mean ± SEM. *P < 0.05; **P < 0.01; ***P < 0.001; ****P < 0.0001. See also Supplementary Figs. 1−3. Source data and exact *p* values are provided as a Source Data file.

**a** Enriched IL-1R pathway in CD34⁺ HSPCs from PMF patients

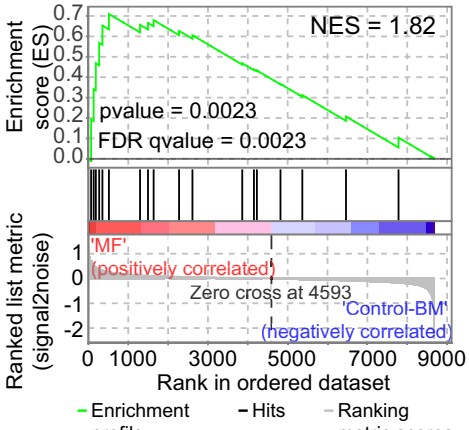

**b** Differential expression of IL-1R pathway genes in CD34⁺ HSPCs from PMF patients

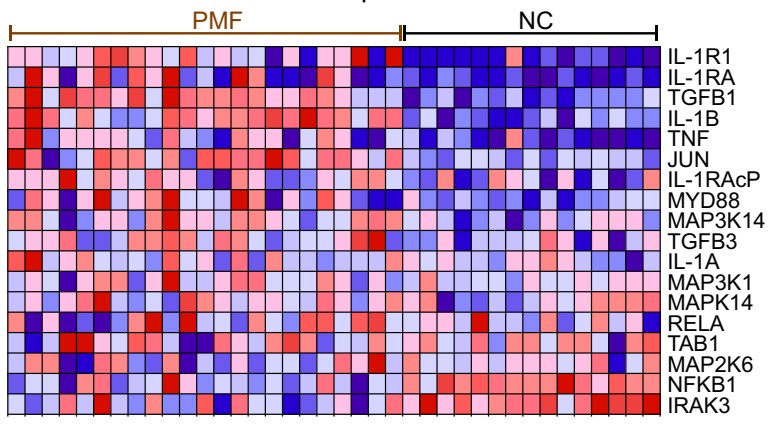

**c** Expression of IL-1 signaling pathway genes in non-transplanted *JAK2-V617F* (*VF*) mice

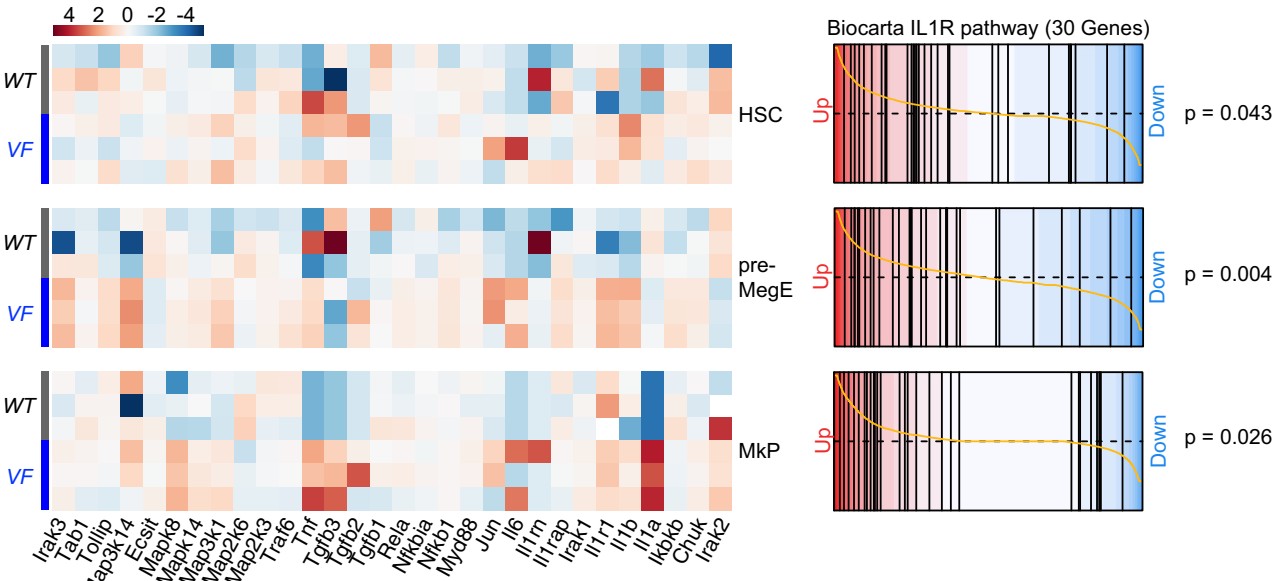

**Fig. 2 | Expression of IL-1 pathway genes are upregulated in MPN. a** Expression of IL-1R pathway gene signatures is tested for enrichment by Gene Set Enrichment Analysis (GSEA) in peripheral blood CD34⁺ HSPCs from PMF patients (*n* = 23) and bone marrow CD34⁺ HSPCs from normal controls (*n* = 15). Comparisons with *p*-value <0.05 and FDR *q*-value <0.05 were considered significant. Analysis of publicly available dataset[19]. Affymetrix data were dowloaded as normalized expression levels from Gene Expression Omnibus database (GSE53482)[19] using the GEOquery package (R, Vienna, Austria. https://www.R-project.org/). The normalization of the expression data was checked by box-plot representation. Gene Set Enrichment Analysis (GSEA) was performed with the GSEAv4.1.0 software (Broad Institute). All gene sets were obtained from GSEA website (https://www.gsea-msigdb.org). Enrichment map was used for visualization of the GSEA results. Normalized Enrichment score (NES) and False discovery rate (FDR) *p*-values were applied after a 10,000 gene set permutations. **b** Heatmap representation of expression levels of IL-1R pathway genes in CD34⁺ HSPCs from PMF patients (*n* = 23) and normal controls (*n* = 15). Analysis of publicly available dataset[19]. **c** Heatmap representation of the differential expression of IL-1 signaling pathway genes in hematopoietic stem cells (HSC; Lin⁻Sca1⁺cKit⁺CD48⁻CD150⁺), megakaryocyte-erythroid precursors (MEP or pre-MegE; Lin⁻Sca1⁻cKit⁺CD41⁻CD16⁻105⁻CD150⁺) and megakaryocyte progenitors (MkP; Lin⁻Sca1⁻cKit⁺CD41⁺CD150⁺) between *VF* (*n* = 3) and *WT* (*n* = 3) mice is shown (left). Barcode plots showing custom Gene Set Enrichment Analysis (GSEA) of the Biocarta IL-1R pathway in HSC, pre-MegE (or MEP) and MkP from *VF* vs *WT* (right). Source data are provided as a Source Data file.

*VF* donors (Fig. 4a and Supplementary Fig. 6a). In line with reduced platelet counts, mice transplanted with bone marrow from *VF;IL-1β⁻/⁻* donors showed reduced frequencies of pre-MegE in bone marrow and spleen (Supplementary Fig. 6b) as well as reduced number of megakaryocytes per visual field in the bone marrow (Supplementary Fig. 7a). Moreover, mice transplanted with *VF;IL-1β⁻/⁻* bone marrow showed significantly reduced TGF-β1 levels in the bone marrow compared to *VF* donors (Supplementary Fig. 7b). IL-1β levels in bone marrow and plasma were low in mice transplanted with *VF;IL-1β⁻/* cells, indicating

the source of IL-1β overproduction in *VF* mice are mainly the *JAK2*-mutant hematopoietic cells (Supplementary Fig. 7c). Spleen weight was slightly reduced in *VF;IL-1β⁻/⁻* recipient mice compared to *VF* (Fig. 4b), and histology of bone marrow revealed reduction in the grade of reticulin fibrosis as well as reduction in the percentage of mice with osteosclerosis (Fig. 4c and Supplementary Fig. 8). Extramedullary hematopoiesis in spleen and liver was decreased and splenic architecture was partially conserved in recipients of *VF;IL-1β⁻/⁻* bone marrow compared to *VF* bone marrow (Supplementary Fig. 6c). No differences

**a** Phenotype of non-transplanted *JAK2*-V617F;*IL-1β^-/-* mice

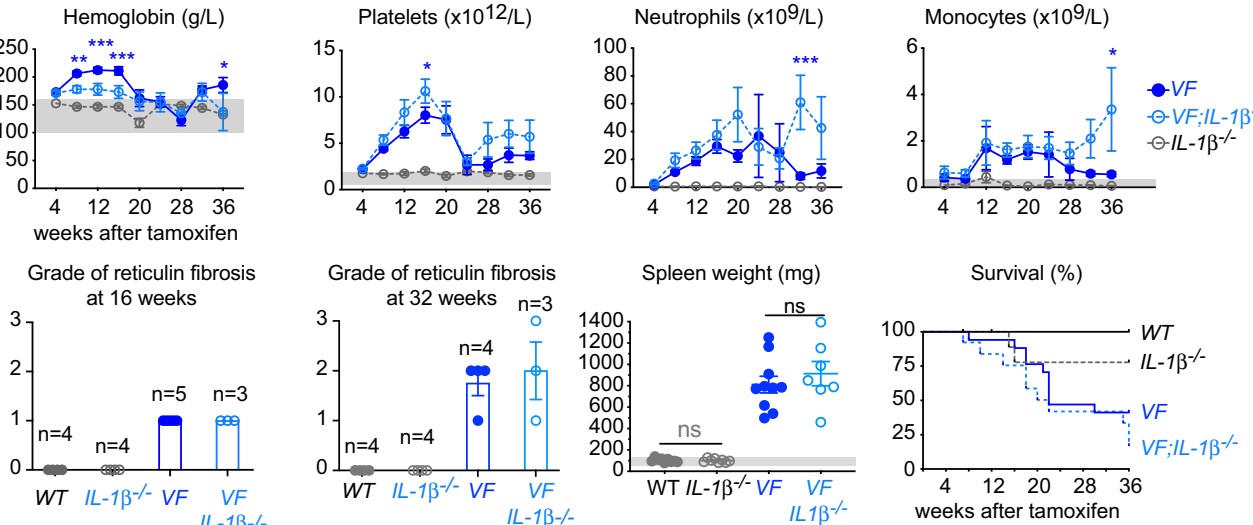

**b** Levels of IL-1 cytokines at 16 weeks after tamoxifen induction

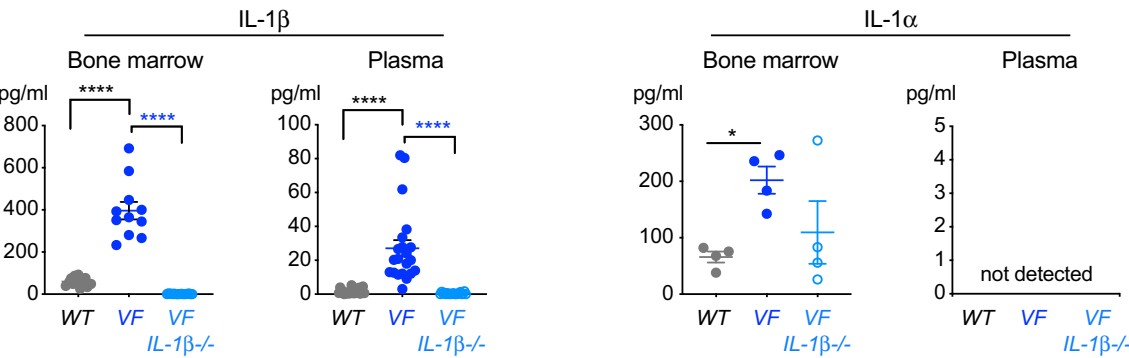

**c** Other pro-inflammatory cytokines levels in *JAK2*-V617F;*IL-1β^-/-* mice 16 weeks after tamoxifen induction

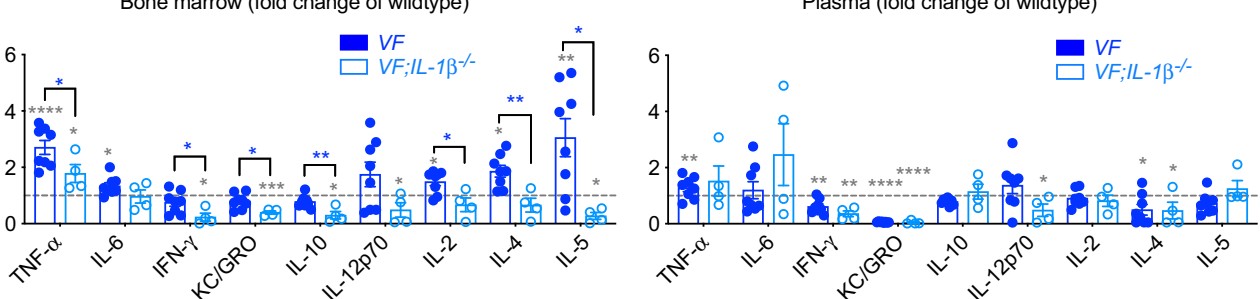

**Fig. 3 | Genetic deletion of *IL-1β* in a *JAK2*-V617F MPN mouse model. a** *Wildtype* (*WT*; *n* = 9), *IL-1β* knock-out (*IL-1β^-/-*; *n* = 11), *Scl;Cre*;V617F (*VF*; *n* = 18) and *Scl;Cre*;V617F; *IL-1β* knock-out (*VF;IL-1β^-/-*; *n* = 13*)* mice were induced with tamoxifen and disease kinetics were followed for 36 weeks. Complete blood counts, grade of reticulin fibrosis at 16- and 32-weeks after tamoxifen and spleen weight at 16 weeks after tamoxifen induction are shown. Kaplan-Meier survival curve showing the percent survival of mice Grey area represents normal range. Two-way ANOVA followed by Tukey's multiple comparison tests were used for multiple group comparisons for blood counts. Two-tailed unpaired t test was performed for spleen weight. **b** left panel: IL-1β protein levels in BM lavage (1 femur and 1 tibia) of *WT*

(*n* = 13), *VF* (*n* = 11) and *VF;IL-1β^-/-* (*n* = 18) and plasma of *WT* (*n* = 21), *VF* (*n* = 21) and *VF;IL-1β^-/-* (*n* = 20) mice at 12–16 weeks after tamoxifen induction. IL-1α levels (middle panel) BM and plasma is shown (*n* = 4 per group). Two-tailed unpaired non-parametric Mann–Whitney t-test was performed. **c** Pro-Inflammatory cytokine levels (normalized to *WT;* dotted line) in BM lavage and plasma of *WT* (*n* = 8), *VF* (*n* = 8) and *VF;IL-1β^-/-* (*n* = 4) mice at 16 weeks after tamoxifen induction. Two-tailed unpaired t-tests were performed for multiple comparisons. Grey asterisk represents the comparison between *WT* and *VF* or *VF;IL-1β^-/-* All data are presented as mean ± SEM. *P < 0.05; **P < 0.01; ***P < 0.001; ****P < 0.0001. See also Supplementary Figs. 4 and 5. Source data and exact p values are provided as a Source Data file.

were observed in *WT* mice transplanted with bone marrow from *IL-1β^-/-* donors versus *WT* donors (Fig. 4 and Supplementary Fig. 6). Histology was also normal in mice transplanted with bone marrow from *IL-1β^-/-* donors or *WT* donors (data not shown).

When *IL-1β^-/-* mice were used as recipients, we observed similar changes in blood counts as in *WT* recipient mice (Fig. 4d). Until week 20, platelet counts in recipients of *VF;IL-1β^-/-* bone marrow were lower than in mice transplanted with *VF* bone marrow and at 16 weeks there

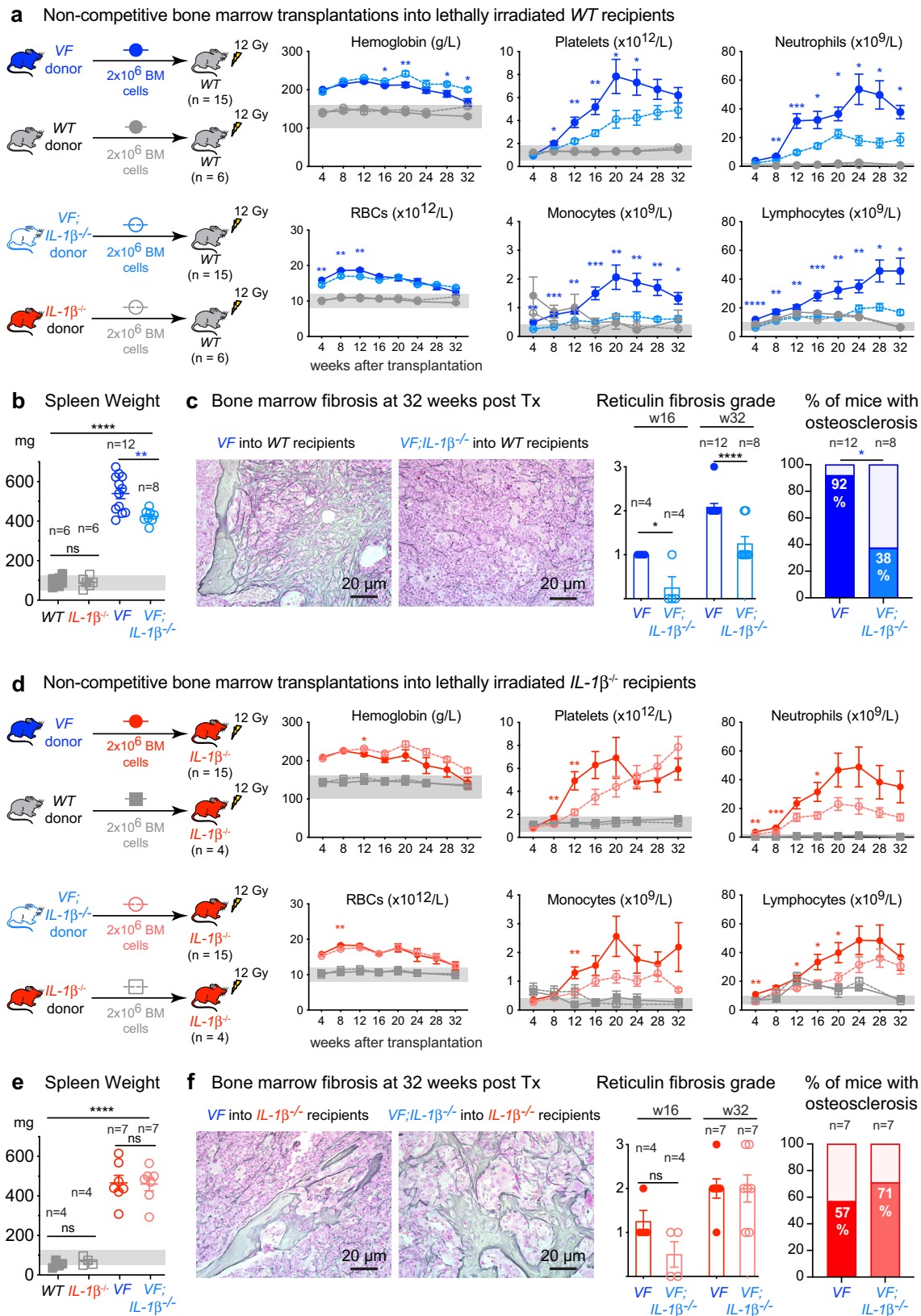

**a** Non-competitive bone marrow transplantations into lethally irradiated *WT* recipients

**b** Spleen Weight

**c** Bone marrow fibrosis at 32 weeks post Tx

**d** Non-competitive bone marrow transplantations into lethally irradiated *IL-1β⁻/⁻* recipients

**e** Spleen Weight

**f** Bone marrow fibrosis at 32 weeks post Tx

was a trend towards lower grade of myelofibrosis (Fig. 4f). However, after 24 weeks the platelet count was higher in *VF;IL-1β⁻/⁻* recipients than in *VF* recipients, and at terminal workup no differences in splenomegaly, grade of reticulin fibrosis or osteosclerosis were detected between the two genotypes (Fig. 4e, f and Supplementary Figs. 6d–f and 8). Thus, at advanced disease stage, *IL-1β⁻/⁻* mice transplanted with

bone marrow from *VF;IL-1β⁻/⁻* mice resembled in phenotype non-transplanted *VF;IL-1β⁻/⁻* mice. While loss of *IL-1β* restricted to hematopoietic cells showed a trend towards increased levels of IL-1Ra in BM, this trend was not observed in mice with complete loss of *IL-1β* in all tissues (Supplementary Fig. 7c), suggesting that the overall activity of IL-1 signaling is reduced when *IL-1β* is lost in hematopoietic cells only.

**Fig. 4 | Loss of *IL-1β* in *JAK2*-V617F mutant cells reduces MPN symptom burden and myelofibrosis. a** Schematic of non-competitive transplantation with 2 million BM cells from tamoxifen induced *VF*, *WT*, *VF; IL-1β–/–*, or *IL-1β–/–* donor mice into lethally irradiated *WT* recipients (*n* = 15 per group). Complete blood counts measured every 4 weeks until 32 weeks after transplantation are shown. Two-tailed unpaired *t*-tests without correction for multiple comparisons was performed. Grey area represents normal range. **b** Bar graph shows the spleen weight at 32 weeks after transplantation. Two-tailed unpaired *t* test was performed. **c** Representative images of bone marrow fibrosis (reticulin fibrosis) are shown at 32-weeks after transplantation. Histological grade of reticulin fibrosis in the BM at 16- and 32-weeks after transplantation is shown in the bar graph. Two-tailed unpaired *t* test was performed. Bar graph showing the percentage of mice with osteosclerosis in the BM. **d** Schematic of non-competitive transplantation with 2 million BM cells from tamoxifen induced *VF*, *WT*, *VF; IL-1β–/–*, or *IL-1β–/–* donor mice into lethally irradiated *IL-1β–/–* recipients (*n* = 15 per group). Complete blood counts measured every 4 weeks until 32 weeks after transplantation are shown. Two-tailed unpaired *t*-tests without correction for multiple comparisons was performed. Grey area represents normal range. **e** Bar graph shows the spleen weight at 32 weeks after transplantation. Two-tailed unpaired *t* test was performed. **f** Representative images of BM fibrosis (reticulin fibrosis) are shown at 32 weeks after transplantation. Histological grade of reticulin fibrosis in the BM at 16- and 32-weeks after transplantation is shown in the bar graph. Bar graph showing the percentage of mice with osteosclerosis in the BM. All data are presented as mean ± SEM. *\*P* < 0.05; *\*\*P* < 0.01; *\*\*\*P* < 0.001; *\*\*\*\*P* < 0.0001. See also Supplementary Figs. 6–8. Source data and exact *p* values are provided as a Source Data file.

## Pharmacological inhibition of IL-1β decreases myelofibrosis in MPN mice

Since the genetic deletion of *IL-1β* in hematopoietic cell showed beneficial effects on myelofibrosis, we also tested the effects of pharmacological inhibition of IL-1β. We used our previously described competitive transplantation model that allows monitoring blood and tissue parameters together with *JAK2* mutant allele burden using a separate *UBC-GFP* reporter transgene that was crossed with the *VF* mice[21]. Bone marrow cells from *VF;GFP* and *WT* donor mice were mixed in 1:1 ratio and transplanted into lethally irradiated recipients (Fig. 5a). The mice developed full PV phenotype with elevated blood counts within 12-16 weeks after transplantation (Supplementary Fig. 9a). Groups of 6 mice were killed at 12, 16, and 20 weeks and the histological grade of reticulin fibrosis was determined (Fig. 5b). At 20 weeks, when all mice within the group displayed myelofibrosis, the remaining mice were randomized and assigned to treatment groups. Anti-mouse IL-1β antibodies, ruxolitinib and combination of both (combo) were well tolerated and did not alter body weight during the 8-week treatment (Fig. 5c). Anti-IL-1β antibody alone reduced platelet and monocyte counts, but prevented the decrease in red cell parameters, whereas ruxolitinib alone had the opposite effects on hemoglobin and platelets (Fig. 5d and Supplementary Fig. 9b). None of the treatments was able to reduce the mutant allele burden in peripheral blood (Fig. 5d) or hematopoietic progenitors in bone marrow and spleen (Supplementary Fig. 9c). Spleen size decreased only in mice treated with ruxolitinib or combo (Fig. 5e).

Vehicle treated mice showed megakaryocytic hyperplasia in bone marrow along with reticulin fibrosis and osteosclerosis (Fig. 5f). Anti-IL-1β antibody reduced reticulin fibrosis as well as the percentage of mice with osteosclerosis, and showed additive effects on both parameters with ruxolitinib (Fig. 5f). IL-1β antibody alone or combination with ruxolitinib almost completely restored splenic architecture and reduced extramedullary hematopoiesis in liver (Supplementary Fig. 9d). Anti-IL-1β antibody treatment did not affect IL-1α levels in BM and plasma (Supplementary Fig. 9e). Anti-IL-1β antibody made IL-1β undetectable in bone marrow and plasma and also reduced the levels of some other pro-inflammatory cytokines (Fig. 5g). The combination with ruxolitinib resulted in even greater suppression of cytokine levels (Fig. 5g). Anti-IL-1β antibody alone reduced the levels of bone marrow TGF-β1 (Supplementary Fig. 10b). *WT* mice transplanted with *WT* bone marrow cells and treated with anti-IL-1β antibody for 8 weeks showed only a minor decrease in lymphocyte numbers, but otherwise no effects on blood counts, spleen weight or bone marrow HSPCs were observed (Supplementary Fig. 10c–g). We confirmed that anti-IL-1β antibody also reduced fibrosis in non-transplanted *VF* mice (Supplementary Fig. 11).

Thus, contrary to the expectations from the complete genetic ablation of *IL-1β* in all tissues, pharmacological inhibition of IL-1β showed beneficial effects on myelofibrosis and course of the disease in *VF* mice comparable to the genetic ablation of *IL-1β* in hematopoietic tissues only.

## Deletion of *IL-1β* in *JAK2*-V617F mutant hematopoietic cells prevents the loss of nestin⁺ mesenchymal stromal cells in bone marrow

*VF* mice were previously shown to display loss of nestin-positive MSCs in bone marrow that was associated with favoring MPN disease manifestations[11]. To test whether loss of *IL-1β* in *JAK2*-mutant donor cells could prevent the destruction of nestin-positive MSCs, we used mice that express a *Nestin-GFP* reporter transgene as recipients for transplantations with bone marrow from *VF*, or *VF;IL-1β–/–*, or *WT* donor mice (Fig. 6a)[24]. Mice transplanted with *VF* bone marrow developed full MPN phenotype (Fig. 6b, c and Supplementary Fig. 12a−c) and showed reduced numbers of *Nestin*-GFP⁺ and *Nestin*-GFP⁺;PDGFRa⁺ MSCs in BM compared to *WT* controls (Fig. 6d, e). In contrast, recipients of *VF;IL-1β–/–* BM showed normal or even increased numbers of *Nestin*-GFP⁺ MSCs in BM (Fig. 6d, e), accompanied by reduced blood counts, in particular normalized platelet counts (Fig. 6b). Importantly, recipients of *VF;IL-1β–/–* BM also showed lower grade of reticulin fibrosis (Fig. 6f), strengthening the link to platelet counts. Consistent with previous observations[11], *VF* mice displayed reduced abundance of Schwann cells and sympathetic nerve fibers in the skull bone marrow (Fig. 6g, h). Importantly, *VF;IL-1β–/–* donor cells failed to inflict damage to neuroglial cells and sympathetic nerve fibers in *Nestin-GFP* recipient mice, and showed Schwann cells and sympathetic nerve fibers densities similar to *WT* controls (Fig. 6g, h). Neuropathy in the skull bone marrow of recipients of *VF* BM correlated with a significant reduction in *Nestin-GFP* + MSC numbers, whereas comparable numbers of *Nestin-GFP* + MSC were observed in recipients of *VF;IL-1β–/–* and *WT* bone marrow (Fig. 6i). We measured the levels of 21 cytokines in bone marrow and plasma, and found increased levels of IL-1β in recipients of *VF* BM, but not in recipients of *VF;IL-1β–/–* or *WT* bone marrow (Supplementary Fig. 12d).

Overall, these results demonstrate that secretion of IL-1β from *JAK2*-mutant BM cells is required to cause neuroglial damage in the BM niche resulting in loss of nestin⁺ MSCs and to maintain high platelet counts. Loss of *IL-1β* in *JAK2*-mutant hematopoietic cells prevented these alterations and correlates with reduced MPN progression to myelofibrosis and osteosclerosis.

## Discussion

IL-1β and IL-1RA levels were elevated in serum of *JAK2*-V617F positive MPN patients (*n* = 120), with mean levels higher in PV and PMF patients than in ET patients. Also consistent with previous studies[4,5], we detected elevated levels of several pro-inflammatory cytokines in MPN patients including IL-8, IL-6, TNF-α, IL-13, IL-10, IL-4, and IL-2. Similar to a recent study in PV patients[25], only IL-1β and IL-1RA showed correlation with *JAK2*-V617F allele burden in peripheral blood (Fig. 1a, b). In addition, we found that the two receptor proteins, IL-1R1 and IL-1RAcP, were expressed at higher levels and in a higher percentage of HSPCs in peripheral blood from MPN patients compared to normal controls (Fig. 1c, d). IL-1R1 and IL-1RAcP were previously shown to be upregulated in acute myeloid leukemia (AML) and chronic myelogenous

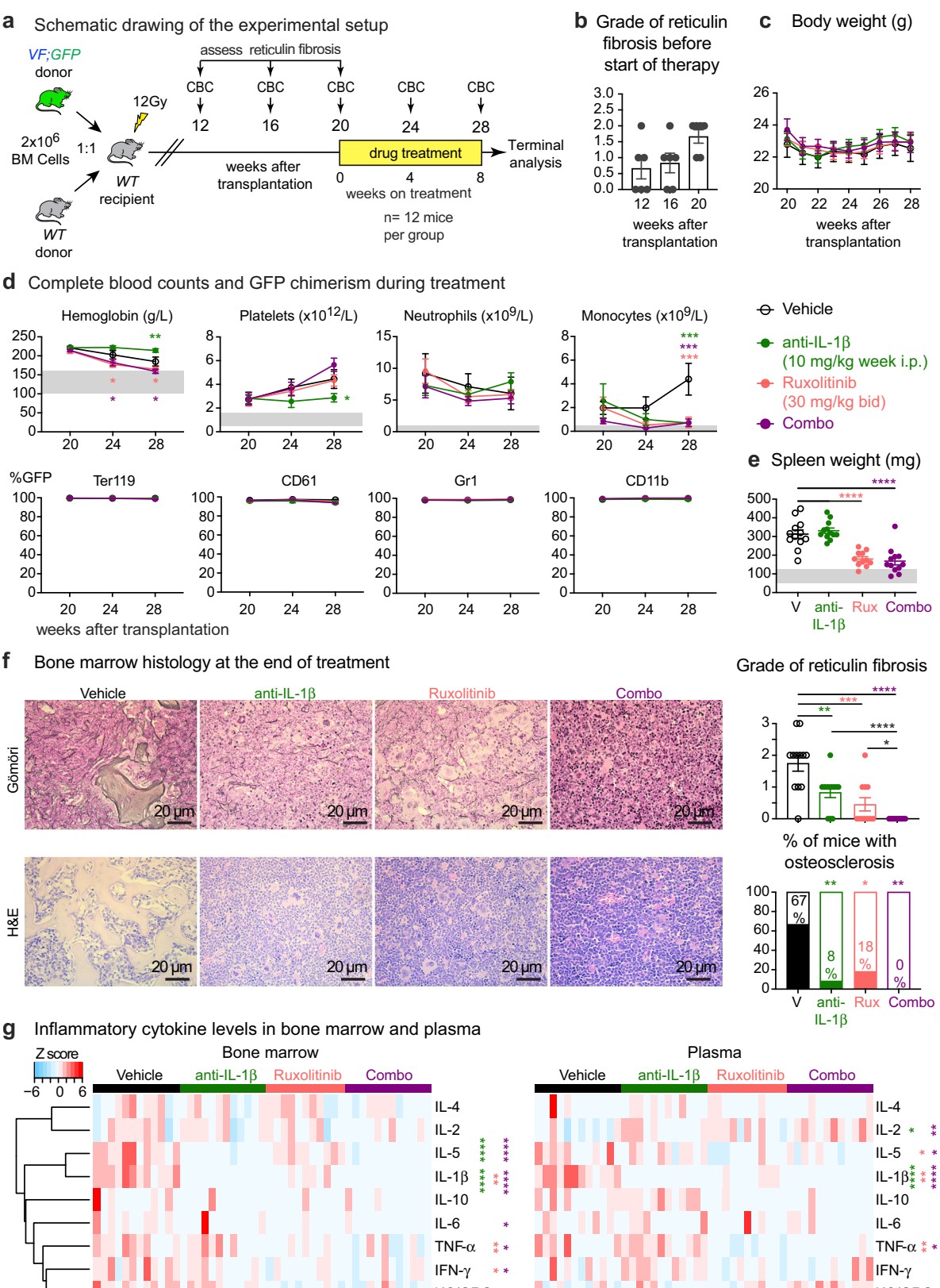

**a** Schematic drawing of the experimental setup

**b** Grade of reticulin fibrosis before start of therapy

**c** Body weight (g)

**d** Complete blood counts and GFP chimerism during treatment

**e** Spleen weight (mg)

**f** Bone marrow histology at the end of treatment

Grade of reticulin fibrosis

% of mice with osteosclerosis

**g** Inflammatory cytokine levels in bone marrow and plasma

leukemia (CML) patients[26–29]. Furthermore, we found a strong correlation between the percentages of IL-1R1+ and IL-1RAcP+ HSPCs and *JAK2*-V617F allele burden, suggesting that *JAK2*-V617F induces expression of IL-1β, which in turn upregulates expression of its own receptors[30,31]. IL-1RAcP is also used by other IL-1 family ligands, including IL-33, IL-36α, IL-36β, and IL-36γ[10], which fits with the higher

percentage of HSPCs that are positive for IL-1RAcP compared to IL-1R1 that is used by IL-1β and IL-1α only. These data are compatible with a model suggesting that upregulating IL-1β is a primary event for the activation of inflammatory signaling in *JAK2*-V617F positive MPN[6,7].

*VF;IL-1β*[−/−] double mutant mice showed reduced levels of proinflammatory cytokines in bone marrow compared to *VF* (Fig. 3c),

**Fig. 5 | Pharmacological inhibition of IL-1β decreased myelofibrosis in MPN mice. a** Experimental setup of the drug treatment. **b** Grade of reticulin fibrosis was determined before therapy in groups of $n = 6$ mice killed at 12-, 16-, and 20-weeks after transplantation. **c** Time course of body weights ($n = 12$ mice per treatment group). Two-way ANOVA followed by uncorrected Fisher's LSD test was performed. **d** Blood counts and mutant cell (% GFP) chimerism in the peripheral blood of vehicle ($n = 12$); ruxolitinib ($n = 11$); anti-IL-1β ($n = 12$); combo ($n = 12$) treated mice in erythroid (Ter119), megakaryocytic (CD61), granulocytic (Gr1), and monocytic (CD11b) lineages. Two-way ANOVA followed by uncorrected Fisher's LSD test was performed. Two-way ANOVA followed by Dunnett's multiple comparisons test was performed for GFP chimerism. **e** Spleen weights of vehicle ($n = 12$); ruxolitinib ($n = 11$); anti-IL-1β ($n = 12$); combo ($n = 12$) treated mice after 8 weeks of drug treatment. Two-tailed unpaired $t$-test was performed. Grey area represents normal range. **f** Representative images of reticulin fibrosis and H&E staining is shown and histological grade of reticulin fibrosis in the BM is illustrated in the bar graph.

Similar results were obtained with other mice in each condition. Stacking bar graph showing the percentage of mice with osteosclerosis in the BM of vehicle ($n = 12$); Ruxolitinib ($n = 11$); anti-IL-1β ($n = 12$); and combo ($n = 12$). Two-tailed unpaired $t$ test was performed for comparisons of fibrosis grades between different groups. $p$ value is computed using Fisher's exact test for presence or absence of osteosclerosis in bone marrow. **g** Heatmap plot showing the inflammatory cytokine levels in the BM lavage and plasma of mice after 8 weeks of drug treatment. Vehicle ($n = 12$); Ruxolitinib ($n = 11$); anti-IL-1β ($n = 12$); combo ($n = 12$). The color bars indicate treatment groups. Heatmap shows $Z$ scores. Two-tailed unpaired $t$-tests without correction for multiple comparisons was performed. Green-colored asterisk is used for comparison of vehicle vs. anti-IL-1β; salmon for vehicle vs. ruxolitinib; plum for vehicle vs. combo. All data are presented as mean ± SEM. *$P < 0.05$; **$P < 0.01$; ***$P < 0.001$; ****$P < 0.0001$. See also Supplementary Figs. 9–11. Source data and exact $p$ values are provided as a Source Data file.

illustrating the role of IL-1β in controlling other cytokines[10]. However, complete loss of *IL-1β* in non-transplanted *VF* mice did not substantially change the course of the disease, in particular, these mice showed no differences in the grade of myelofibrosis (Fig. 3a). *VF;IL-1β*[−/−] mice showed only slightly reduced red cell parameters, and overall no significant changes in platelet and leukocyte counts compared to *VF*. In contrast, transplantation of *VF;IL-1β*[−/−] bone marrow into *WT* recipients resulted in decreased platelet counts, reduced infiltration of megakaryocytes in BM and spleen and reduced degree of myelofibrosis and osteosclerosis (Fig. 4a–c). Also *IL-1β*[−/−] recipients transplanted with *VF;IL-1β*[−/−] bone marrow as expected initially displayed lower platelet counts and at 16 weeks showed a trend towards lower grade of MF (Fig. 4). However, after 24 weeks the platelets increased in these *VF;IL-1β*[−/−] transplanted mice to levels higher than in the *VF*-transplanted group and at 32 weeks there was no difference in the grade of myelofibrosis between the two groups. Thus, transplantation of *VF;IL-1β*[−/−] bone marrow into *IL-1β* deficient recipients, similar to complete *IL-1β* knockout, abolished the beneficial effects on myelofibrosis and osteosclerosis. Nevertheless, anti-IL-1β antibody alone effectively eliminated IL-1β protein, reduced platelet counts and also reduced the grade of myelofibrosis compared with vehicle (Fig. 5 and Supplementary Fig. 11). The reasons for the discrepancies in phenotypes between loss of *IL-1β* in hematopoietic tissues only versus the complete genetic loss of *IL-1β* and the effects of anti-IL-1β antibody remain unclear, but might be partly explained by developmental compensation when IL-1β is deleted throughout life. The beneficial effects on myelofibrosis and osteosclerosis in our mouse models strongly correlated with the reduction in platelet counts in the various settings of inhibiting or deleting *IL-1β*.

The mechanism by which IL-1β promotes myelofibrosis involves direct effects on megakaryopoiesis and on bone marrow microenvironment. Platelets and megakaryocytes have been shown to be prime drivers in the pathogenesis of myelofibrosis[32,33]. IL-1β has a direct positive effect on megakaryopoiesis[34–37] and promotes polyploidisation of megakaryocytes through NFκB and MAPK signaling[38]. TGFβ release from platelets has been implicated as a key mediator of the pro-fibrotic process[39]. Our data is in-line with these studies, as we also observed increased TGFβ1 serum levels in MPN patients (Supplementary Fig. 3) and reduced TGFβ1 levels in bone marrow of *VF* mice with genetic or pharmacological inhibition of IL-1β (Supplementary Figs. 7b and 10b). Furthermore, we show that deleting IL-1β in *VF* hematopoietic cells prevented neuropathy, i.e. damage inflicted on the Schwann cells and sympathetic nerve fibers[11]. As a consequence, nestin-positive MSCs were preserved in bone marrow of *VF;IL-1β*[−/−] mice and the presence of nestin-positive MSCs correlated with reduced myelofibrosis (Fig. 6). A link between increase of nestin+ MSCs and reduced myelofibrosis was also found in our clinical phase II trial using Mirabegron, a β−3 sympathomimetic

agonist, that corrected the damage inflicted by the MPN clone on the nestin+ MSCs[40].

Anti-IL-1β antibody completely neutralized IL-1β in bone marrow and plasma of *VF* mice and substantially reduced reticulin fibrosis and osteosclerosis (Fig. 5). This antibody is selective for IL-1β and does not bind IL-1α (Supplementary Fig. 9e). The half-life of this anti-IL-1β antibody (01BSUR) in mice was >300 h (Novartis internal data), and has shown efficacy in several pre-clinical models[41–43]. Anakinra, a recombinant form of naturally occurring human IL-1Ra partially ameliorated disease phenotype in *VF* mice, but showed no effect on myelofibrosis[11]. The half-life of Anakinra is only 4–6 h and an up to1000-fold molar excess over IL-1β is required to completely block IL-1 signaling[10]. Ruxolitinib alone only partially decreased IL-1β levels and IL-1β has been shown to act on cells even at picomolar concentrations[14,44]. Ruxolitinib alone showed variable effects on fibrosis in preclinical mouse models. The effects were mainly dependent on the dose and duration of treatment. Higher doses of ruxolitinib (60–90 mg/kg, BID) or longer treatment regimen (more than 3 weeks) were effective in reducing myelofibrosis in mice[7,45], whereas, shorter treatment and/or lower dose (30 mg/kg; 3 weeks, QD) did not reduce myelofibrosis[46]. We used ruxolitinib at 30 mg/kg BID, but we treated for 8 weeks. Thus, our results are comparable with the other reports. The anti-IL-1β antibody showed additive effects with ruxolitinib in reducing myelofibrosis, osteosclerosis as well as inflammatory cytokines in MPN mice. This is in line with a recent report, which showed that inhibiting the NF-kB inflammatory pathway using a BET-inhibitor in combination with ruxolitinib was more effective than monotherapy in reducing myelofibrosis in a MPN mouse model[7].

Using genetic and pharmacological approaches, we show that IL-1β inhibition reduced myelofibrosis in a preclinical *JAK2*-V617F MPN mouse model. Furthermore, the combination therapy with Jak1/2 inhibitor resulted in complete reversal of myelofibrosis and osteosclerosis. Our data highlight the role of IL-1β in MPN disease progression to myelofibrosis and provide a rationale for a clinical trial with anti-IL-1β antibody in MPN.

## Methods
### MPN patients
Blood samples and clinical data of MPN patients were collected at the University Hospital Basel, Switzerland. The study was approved by the local Ethics Committees (Ethik Kommission Beider Basel). Written informed consent was obtained from all patients in accordance with the Declaration of Helsinki. The diagnosis of MPN was established according to the 2016 revision of the World Health Organization classification of myeloid neoplasms and acute leukemia[47]. Information on diagnosis, progression, and gene mutations are specified in Supplementary Data 1.

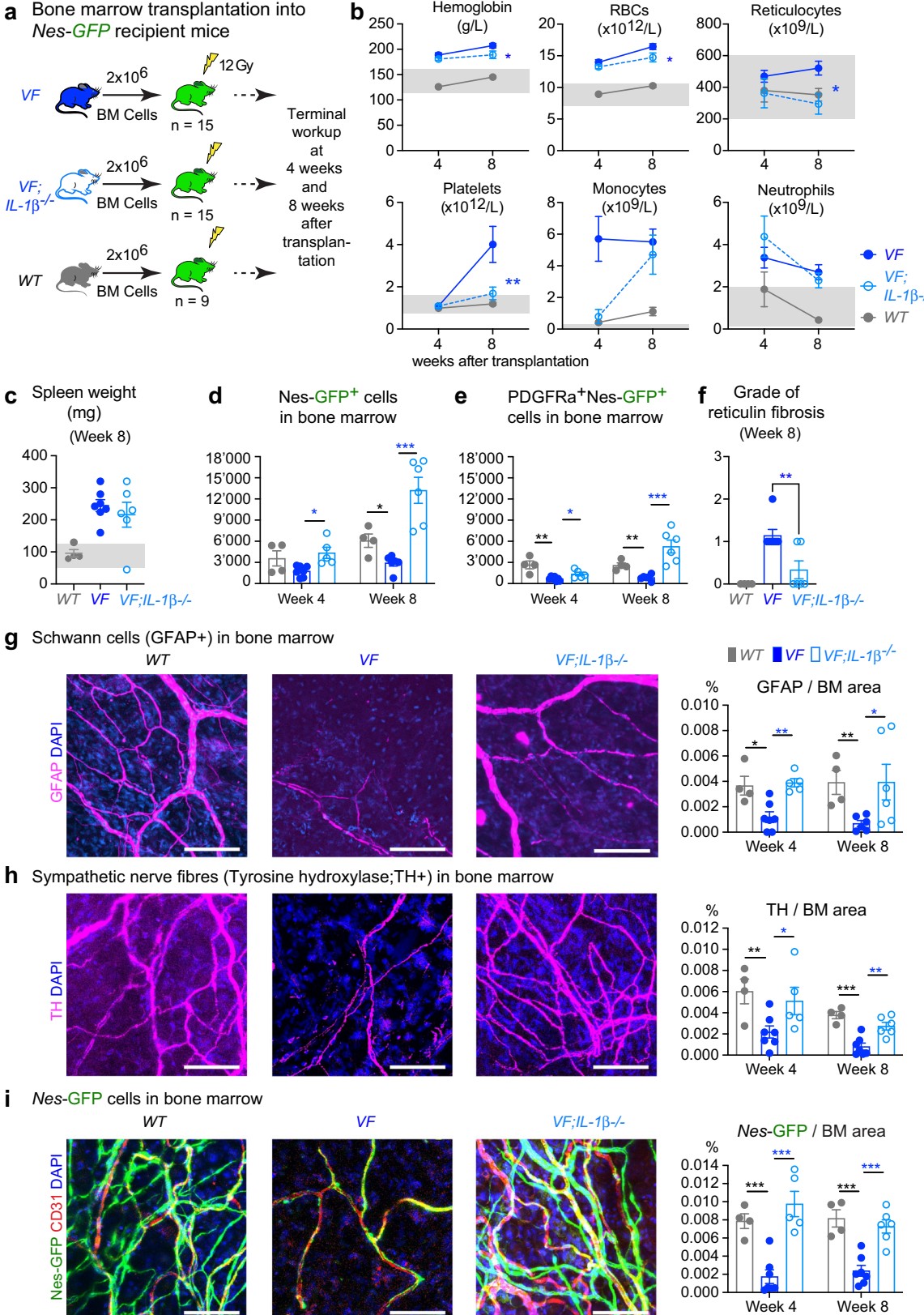

### Quantification of *JAK2*-V617F variant allele frequency in genomic DNA

DNA from granulocytes was prepared using the QIAamp DNA Mini Kit using manufacturer's instructions. An allele-specific polymerase chain reaction (AS-PCR) was performed for the detection of *JAK2*-V617F in genomic DNA[48]. PCR amplification was performed forward primer 5′-GTTTCTTAGTGCATCTTTATTATGGCAGA-3′ and reverse primers 5′−6 Fam-AAATTACTCTCGTCTCCACAGAA-3′ and 5′−6Fam-TTACTCTCGT CTCCACAGAC-3′. The amplicons generated by AS-PCR were analyzed by fragment analysis with ABI3130xl Genetic Analyzer (Applied Bio-systems Inc.). The mutant allele burden was calculated by Peak height$_{mut}$/(Peak height$_{mut}$ + Peak height$_{wt}$) × 100%.

**Fig. 6 | Deletion of *IL-1β* in *JAK2*-V617F mutant hematopoietic cells prevented the loss of nestin⁺ MSCs in bone marrow. a** Scheme of non-competitive (1:0) transplantation into *Nestin-GFP* mice. **b** Complete blood counts at 4-weeks (*VF*; *n* = 7, *VF;IL-1β⁻/⁻*; *n* = 6, and *WT*; *n* = 4) and 8-weeks (*VF*; *n* = 7, *VF;IL-1β⁻/⁻*; *n* = 6, and *WT*; *n* = 4) after transplantation. **c** Spleen weights after 8 weeks of transplantation (*VF*; *n* = 7, *VF;IL-1β⁻/⁻*; *n* = 6, and *WT*; *n* = 4). **d** Number of Ter119⁻CD45⁻CD31⁻GFP⁺ cells in BM (1 tibia and 2 hip bones) at 4-weeks (*VF*; *n* = 7, *VF;IL-1β⁻/⁻*; *n* = 5, and *WT*; *n* = 4) and 8-weeks (*VF*; *n* = 6, *VF;IL-1β⁻/⁻*; *n* = 6, and *WT*; *n* = 4) after transplantation. **e** Total number of Ter119⁻CD45⁻CD31⁻GFP⁺ cells co-expressing platelet derived growth factor receptor α (PDGFR α) at 4-weeks (*VF*; *n* = 7, *VF;IL-1β⁻/⁻*; *n* = 5, and *WT*; *n* = 4) and 8-weeks (*VF*; *n* = 6, *VF;IL-1β⁻/⁻*; *n* = 6, and *WT*; *n* = 4). **f** Grade of reticulin fibrosis 8-weeks after transplantation. *VF*; *n* = 7, *VF;IL-1β⁻/⁻*; *n* = 6, and *WT*; *n* = 4. **g** Representative images of glial fibrillary acidic protein (GFAP)-positive Schwann cells in skull BM and quantification of GFAP area at 4-weeks (*VF*; *n* = 7, *VF;IL-1β⁻/⁻*; *n* = 5, and *WT*; *n* = 4) and 8-weeks (*VF*; *n* = 6, *VF;IL-1β⁻/⁻*; *n* = 6, and *WT*; *n* = 4) after transplantation (right). **h** Representative images of tyrosine hydroxylase (TH)-positive sympathetic nerve fibers in skull BM and quantification at 4-weeks (*VF*; *n* = 7, *VF;IL-1β⁻/⁻*; *n* = 5, and *WT*; *n* = 4) and 8-weeks (*VF*; *n* = 7, *VF;IL-1β⁻/⁻*; *n* = 6, and *WT*; *n* = 4) after transplantation (right). **i** Representative images of *Nestin*-GFP cells in skull BM and quantification at 4-weeks (*VF*; *n* = 7, *VF;IL-1β⁻/⁻*; *n* = 5, and *WT*; *n* = 4) and 8-weeks (*VF*; *n* = 7, *VF;IL-1β⁻/⁻*; *n* = 6, and *WT*; *n* = 4) (right). Similar results were obtained with other mice of each genotype in **g**–**i** (left panel). Scale bar is 100 μm in **g**–**i** (left panel). Statistical significances in all graphs were determined by multiple unpaired two-tailed *t*-tests. Grey area represents normal range. All data are presented as mean ± SEM. *$P < 0.05$; **$P < 0.01$; ***$P < 0.001$; ****$P < 0.0001$. See also Supplementary Fig. 12. Source data and exact *p* values are provided as a Source Data file.

## qPCR

*IL1B (IL-1β)*, *IL1RN (IL-1RA)*, *CASP1 (caspase1)* and *ACTB (β-actin)* gene expression in human granulocytes were quantified by TaqMan gene expression assay (Assay ID: Hs01555410_m1, Hs00893626_m1, Hs00354836_m1 and Hs01060665_g1; ThermoFisher Scientific). Each sample was run in triplicates using 25 ng cDNA in a 384 well plate and the qPCR was performed using VIIA 7 real time PCR instrument from Applied Biosystems.

## Transgenic mice

Tamoxifen inducible *SclCre^ER;V617F (VF)* mice were described previously[20]. *VF* mice were crossed with *UBC-GFP* strain[49], and bone marrow (BM) cells from *VF;GFP* mice that co-express GFP as a reporter were used for competitive transplantations. Double mutant *VF;IL-1β⁻/⁻* mice were generated by crossing *VF* mice with previously described mice lacking *IL-1β*[50]. We used *Nestin-GFP* reporter mice as transplantation recipients[24]. Cre-recombinase expression in transgenic mice was induced by intraperitoneal (i.p.) injections of 100 μg/g body weight tamoxifen (Sigma Aldrich) for 5 consecutive days. All mice were of pure C57BL/6N background, and kept under specific pathogen-free conditions with free access to food and water in accordance to Swiss federal regulations. All animal experiments were approved by the Cantonal Veterinary Office of Basel-Stadt, Switzerland.

## Bone marrow transplantations

Transplantations were performed with BM cells harvested from transgenic mice 6–8 weeks after induction with tamoxifen. For non-competitive transplantation assays, erythrocyte depleted total BM cells (2 million) isolated from C57BL/6 (*WT*), *VF* or *VF;IL-1β⁻/⁻* donor mice were transplanted into lethally irradiated *WT*, *IL-1β⁻/⁻*, or *Nestin-GFP* recipients. The *WT* recipients were purchased from Janvier Labs. Blood samples were taken from lateral tail vein every 4–6 weeks to measure complete blood counts (CBC). CBC were determined using an Advia120 Hematology Analyzer using Multispecies Version 5.9.0-MS software (Bayer). In the final phase of the experiment, the recipients were euthanized by $CO_2$ asphyxiation and the tissue/blood samples were taken for further analysis.

## Flow cytometry

Frozen PBMCs from MPN patients and normal controls were thawed and stained after blocking Fcγ receptors (#564220, BD) with following human antibodies: lineage-FITC (1:20; 348701), CD34-Pacific Blue (1:100; 343512), CD38-APC (1:50; 356606), CD123-BV605 (1:100; 306026), and CD41-PE-Cy5 (1:50; 343512) from BioLegend, CD45RA-BV786, (1:50; 563870; BD biosciences) and IL-1R1-PE (1:20; FAB269P), IL-1RAcP-PE (1:20; FAB676P) or isotype goat IgG-PE antibody (1:20; IC108P) from R&D systems. Mouse BM cells were harvested from long bones (2 tibias and 2 femurs) by crushing bones with mortar and pestle using staining media (Dulbecco's PBS + 3% FCS + pen/strep). Cells were filtered through 70 μm nylon mesh to obtain a single-cell suspension.

Total spleen cells were harvested by crushing the spleen against 100 μm cell strainer. Red blood cells were lysed (ACK buffer, Invitrogen) and stained with following antibodies for FACS analysis: a mixture of biotinylated monoclonal antibodies CD4 (1:200; 100404), CD8 (1:200; 100704), B220 (1:200; 103204), TER-119 (1:100; 116204), CD11b (1:400; 101204), and Gr-1(1:400; 108404) from BioLegend was used as the lineage mix (Lin) together with Sca-1-APC-Cy7 (1:100; 108126), CD117 (c-kit)-BV711 (1:100; 105835), CD48-AF700 (1:100; 103426), CD150 (SLAM)-PE-Cy7 (1:100; 115914), CD16-PE (1:100; 101308), CD41-BV605 (1:100; 133921), CD105-PerCP-Cy5.5 (1:100; 120416) from BioLegend and CD34-AF647 (1:25; 560230; BD biosciences) were used as primary antibodies. Cells were washed and stained with streptavidin pacific blue secondary antibody (Invitrogen). Mouse stromal cells were obtained by crushing mouse bones directly in 0.25% collagenase I in 20% FBS/PBS solution and digesting bones and cells at 37 °C water bath for 45 min. Cells were filtered through 70μm nylon mesh, red blood cells were lysed (ACK buffer, Invitrogen) and cells were stained with CD45-PE-Cy7 (1:100; 103114), CD31-PerCP-Cy5.5 (1:100; 102420), TER-119-APC (1:200; 116212), Sca-1-APC-Cy7 (1:100; 108126), PDGFRα-PE (1:100; 135906) from BioLegend. Sytox-Blue or Green (Invitrogen) was used to exclude dead cells during FACS analysis. Live, singlet cells were selected for gating. Cells were analyzed on a Fortessa Flow Cytometer (BD biosciences). Data were analyzed using FlowJo (version 10.7.1) software.

## Cytokine analysis

Mouse blood was collected in EDTA tubes by cardiac puncture and platelet depleted plasma was collected by centrifuging the blood at 5000×*g* for 20 min at 4 °C. Mouse BM lavage was collected by flushing one femur and one tibia with 500 μl PBS and centrifuging the cell suspension at 300×*g* for 10 min at 4 °C. IL-1β and other pro-inflammatory cytokine levels in mouse BM and plasma as well as in human serum were measured by ELISA kits from R&D systems and Mesoscale Discovery according to manufacturer's instructions. Single analyte data was plotted in GraphPad Prism software using an XY data table and the standard curve was analyzed using a sigmoidal 4-PL equation and the values of unknowns were interpolated. Multiplex cytokine data from a 96-well plate was read using Mesoscale Meso Sector S 600 instrument and the data was analyzed with Discovery Workbench 4.0 software. The cytokine data were then normalized by *Z* score transformation using the scale () function in R and visualized with the heatmap.2 function of the gplots package.

## Histology

Bones (sternum and/or femur), spleens and livers were fixed in formalin, embedded in paraffin and sectioned. Tissue sections were stained with H&E and Gömöri for the analysis of reticulin fibers. Pictures were taken with ×10, ×20, and ×40 objective lens using Nikon Ti inverted microscope and NIS Software. For staining nerve fibers and

Schwann cells, mouse skull bones were fixed in 2% formaldehyde/PBS solution for 2 h at 4 °C, washed with PBS and stored in PBS at 4 °C until further analysis. The whole mount immunostaining of the skull bones was performed and antibodies used for immunofluorescence staining were anti-TH (Rabbit pAb, Millipore) and anti-GFAP (Rabbit pAb, Dako). Confocal images were acquired with a laser scanning confocal microscope (Zeiss LSM 700). At least 3 different sections were used for quantification using ImageJ software.

## Pharmacological treatments in vivo

BM cells from *VF;GFP* donor mice were mixed with BM cells of *WT* in 1:1 ratio (2 million cells) and transplanted into lethally irradiated (12 Gy) *WT* female recipients. Mice (*n* = 6) selected randomly from the cohort and killed at week-12, −16, and −20 post-transplant to assess the grade of reticulin fibrosis in BM. At 20-weeks post-transplant, mice were randomized into following 4 groups of 12 mice each and treated for 8-weeks: (1) Vehicle: 0.5% methylcellulose (oral gavage) + isotype antibody (10 mg/kg*qw, i.p.), (2) Ruxolitinib (30 mg/kg*bid, oral gavage) + isotype antibody (10 mg/kg*qw, i.p.), (3) mouse IgG2a anti-mouse IL-1β antibody[41–43] (01BSUR) (10 mg/kg*qw, i.p.) + 0.5% methylcellulose (oral gavage), and (4) Combination of ruxolitinib (30 mg/kg*bid, oral gavage) and anti-mouse IL-1β antibody (10 mg/kg*qw, i.p.). Mouse IgG2a isotype control antibody, anti-mouse IL-1β antibody and ruxolitinib phosphate salt were supplied by Novartis Pharma AG (Basel, Switzerland).

## Isolation of RNA and RNA sequencing analysis

RNA-Seq analysis was performed as previously described[51,52]. Briefly, RNA from FACS sorted long-term hematopoietic stem cell (HSC; Lin⁻Sca1⁺cKit⁺CD48⁻CD150⁺), megakaryocyte-erythroid precursors (MEP or pre-MegE; Lin⁻Sca1⁻cKit⁺CD41⁻CD16⁻105⁻CD150⁺) and megakaryocyte progenitors (MkP; Lin⁻Sca1⁻cKit⁺CD41⁺CD150⁺) from bone marrow were prepared using Picopure RNA isolation kit (Applied Biosystems). The quality and concentration of total RNA was determined on Agilent 2100 Bioanalyzer using the Eukaryote Total RNA Pico Assay (RNA Index number >7 was used for quality check). RNA was reverse transcribed and cDNA amplified with SMART-Seq v2 or v4 (Takara). Libraries were prepared with Nextera XT (Illumina) according to manufacturer's instructions. Samples were pooled to equal molarity and run on the Fragment Analyzer for quality check and used for clustering on the NextSeq 500 instrument (Illumina). Samples were sequenced using the NextSeq 500 High Output kit 75-cycles (Illumina), and primary data analysis was performed with the Illumina RTA version 2.1.3 and bcl2fastq-2.16.0.10.

## Statistical analyses

Blood count and organ weights of mice were recorded as indicated in figure legends. Histological staining from sternum/femur, spleen, and liver was analyzed by a pathologist. The number of animals and replicates can be found in the respective figure legends. The unpaired two-tailed Student's *t*-test analysis was used to compare the mean of two groups. Normality tests were performed to test whether the data follows a normal distribution. When the distribution was not normal, non-parametric Mann−Whitney *t*-tests were performed. For samples with significantly large variances, Welch's correction was applied for *t*-test. Two-tailed unpaired multiple *t*-tests with or without correction were also performed for the comparison of multiple groups or one-way or two-way ANOVA analyses followed by Dunn's, Tukey's or Bonferroni's multiple comparison tests were used for multiple group comparisons. Survival rate in mouse experiments was represented with Kaplan-Meier curves and significance was estimated with the log-rank test. Data were analyzed and plotted using Prism software version 7.0 (GraphPad Inc.). All data are represented as mean ± SEM. Significance is denoted with asterisks (*$p < 0.05$, **$p < 0.01$, ***$p < 0.001$, ****$p < 0.0001$).

## Reporting summary

Further information on research design is available in the Nature Research Reporting Summary linked to this article.

## Data availability

RNAseq dataset used in this study are available at GEO database with accession numbers GSE132570 and GSE116571. All other data that support the findings of this study are available within the article, its supplementary information, or Source Data file. Source data are provided with this paper.

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

## Acknowledgements

This work was supported by grants from the Swiss National Science Foundation (31003A_166613 and 310030_185297) and Swiss Cancer Research (KFS-3655-02-2015 and KFS-4462-02-2018) to R.C.S, National Health Service Blood and Transplant (United Kingdom), European Union's Horizon 2020 research (ERC-2014-CoG-648765), MRC-AMED grant MR/V005421/1 and a Programme Foundation Award (C61367/A26670) from Cancer Research UK to S.M.-F. The authors thank Marc Donath, Marianne Böni, and members of their laboratory for helpful discussions, Rao N. Tata for advice during the initial phase of the project, and members of our laboratory for critical reading of the manuscript. We thank the Bioinformatics core facility, Animal core facility and Flow cytometry core facility of Department of Biomedicine for excellent technical support.

## Author contributions

S.R. designed and performed the research, analyzed data, and wrote the manuscript; E.G. performed research and analyzed data; N.H., D.L.P., C.B.S., H.H.S., and G.M.S. performed research; S.D. performed and analyzed histopathology of mouse tissues; C.J.F. analyzed data; S.M.F.

designed research and analyzed data; R.C.S. designed the research, analyzed the data, and wrote the manuscript.

## Competing interests

R.C.S. has consulted for and received honoraria from Novartis and Celgene/BMS, he is a scientific advisor/SAB member and has equity in Ajax Therapeutics; N.H. owns stocks in the company Cantargia; C.J.F. is a full-time employee of Novartis Pharma AG. The inhibitor studies were carried out in the laboratory of R.C.S. with inhibitors provided by Novartis. The remaining authors declare no competing financial interests.
