## [Peer Review File · Nature Communications]

Inhibition of interleukin-1 β reduces myelofibrosis and osteosclerosis in mice with JAK2-V617F driven myeloproliferative neoplasmREVIEWER COMMENTS

Reviewer #1 (Remarks to the Author): with expertise in haematopoiesis and inflammatory cytokines

The authors show that IL-1b is elevated in MPN patients with JAK2 mutations as well as in a JAK2 mutant MPN mouse model. Genetic knock out of IL-1b from the hematopoietic stem and progenitor cells reduced bone marrow fibrosis as well as osteosclerosis in the MPN mice. However, this effect was not seen in the mice with IL-1b knockout in the niche. Moreover, the authors show that pharmacological inhibition of the IL-1b receptor in combination with Ruxolitinib reduces bone marrow fibrosis and osteosclerosis. The manuscript is very well written. Overall, further mechanistic insight and inclusion of key controls is needed.

MAJOR COMMENTS:

- Figure 1C-D and Suppl Fig 1B: In the supplementary figure, the authors should consider including data for the number/percentage of just HSCs and HSPCs to get a baseline read if there are differences in just stem and progenitor cells between NC and MPN patients, before going into the IL1R1+ subset of the HSPCs/HSCs.
- Figure 3: The authors should include healthy WT cells as donors for their control transplants in both the competitive WT recipients and non-competitive IL1b^{-/-} recipients.
- Figure 4D: the authors should include at minimum in the extended figures the data demonstrating the effect of IL1b blockade treatment on the WT donor cells and not just the MPN.
- The authors should provide some mechanistic insight as to how IL1b inhibition is reducing fibrosis in order to extend the manuscript
- In the text the authors claim, "Taken together, these results demonstrate that IL-1 β deficiency in this MPN mouse model reduced inflammation in the bone marrow, but apart from slightly increasing platelet and leukocyte numbers while decreasing red cell parameters, did not affect the overall course of MPN disease." However, these data argue against the hypothesis because there are a similar number of cells in the BM and Spleen although lowered inflammation. Is inflammation associative or causative in MPN progression?
- "While loss of IL-1 β restricted to hematopoietic cells increased the levels of IL-1Ra in BM, the complete loss of IL-1 β in all tissues was not accompanied by an increase in IL-1Ra (Supplementary Figure S5), suggesting that the overall activity of IL-1 signaling is reduced when IL-1 β is lost in hematopoietic cells only." While the increase may be trending, it is not significant. These data seem to detract from the main point of the manuscript and the authors write in the discussion that there isn't a good explanation for them. It may be best to exclude the data from the manuscript to retain focus.

MINOR COMMENTS:

- Text edit (italicized). Overall, these results show a positive correlation between JAK2-V617F and increased IL-1 signaling in MPN patients.
- Text edit (italicized). IL-1 β is considered a master regulator of inflammation that controls the production of multiple pro-inflammatory cytokines and induces its own expression via a positive feedback loop in an autocrine or paracrine manner.

Reviewer #2 (Remarks to the Author): with expertise in myeloproliferative neoplasms

Rai, Skoda et al. dissect the role of IL1b in patient samples and murine models on JAK2V617F-driven MPN associated with bone marrow fibrosis. IL1b as a key immunoregulatory and proinflammatory cytokine, is produced by the inflammasome and has gained significant attention in the pathogenesis of the myeloid neoplasms in the last couple of years. The authors convincingly show that IL1b is upregulated in human disease and also show an association with JAK2 mutated disease (in their own data set and using publicly available data sets). In particular the data on patient samples is strong and detailed and really raises the question how knockout of IL1b affects the MPN phenotype and the development of bone marrow fibrosis. Given the title of the article, the data presented in the manuscript were rather surprising as one would have expected an amelioration of the MPN

phenotype. However, the results using the genetic knockout in HSCs and stromal cells (recipient mice after transplant) show rather subtle effects. In my opinion this is still very interesting but it should be highlighted that the effect is rather minor and most effect is seen in the knockout in HSCs after transplant (maybe after addition of additional stress).

The results of the anti-IL1b inhibitor are interesting but also here it seems like the major effect is caused by Ruxolitinib and the combinatorial effect is only minor. More mechanistic insights on the link between JAK-STAT signaling, the inflammasome and IL1b will help to understand the findings. Recent work in solid tumors suggested a link between IL1, JAK-STAT and TGFb - it would be highly interesting to analyse this mechanistic link in the context of experiments in this paper.

More specifically, I have the following concerns/comments:

The patient data in figure 1 are convincing. Figure 1f using the publicly available data sets indicates an interesting link between TGFb and IL1, at least they seem to change in the same direction. Can you follow up on this potential correlation?

In general, all headlines in the manuscript summarize the main finding of the paragraph. However, on page 6, the headline rather states the approach but not the finding. This is most likely due to the fact that there is no effect of the genetic KO of IL1b on the phenotype. I was really surprised to see the results and also had to look at the figure multiple time as the statements are pretty weak instead of summarizing this as a negative result, which is also an interesting finding (but unexpected based on the title, abstract etc).

Figure 2 is also far too dense and should be focused on the main findings. 2a is composed of 13 (!) graphs. This figure should be reduced to summarize the main message. The body weights can go into the supplements and the authors should consider showing spleen/body weight.

Figure 2b is also composed of too many graphs. It is interesting that although IL1b is reduced in the KO, there is basically no effect on the phenotype. The legend for 2d is missing - please also only focus on significant results in this graph for better readability

It would have been good to have a wild-type control in the experiment in 2a to have the baseline hematopoietic phenotype after transplant.

Please show also representative images of endosteal areas for the IL1b knockout in 3c - this is the area that's affected first by fibrosis and is not shown here (not representative, comparable).

It is confusing that the grade of reticulin fibrosis in VF is 2, but 92% of mice had osteosclerosis? How was this calculated?

It is interesting that there is no stromal/recipient KO of IL1b emphasizing that the effect is mainly on HSCs as also the patient samples in 1 would have suggested. Did the authors ever check for a correlation of IL1b to the fibrosis grade (in all their samples independent of the MPN entity).

The representation of fibrosis grades in 4b is suboptimal, please improve.

The most striking finding in the blood counts after inhibitor treatment seemed to be on monocytes. Did the authors further look into this?

It is confusing that the inhibitor does not have an effect of spleen weight but the reticulin grade is reduced. How do the authors explain this? The megakaryocytes in the anti IL1b treatment still look very abnormal. Who quantified the fibrosis grade? Was this blinded? Multiple reviewers?

Were the reticulin images in 4f taken in comparable locations of the bone?

In the quantification of cytokines, the major effect seems to be caused by Ruxolitinib not IL1b inhibition. What is the cellular source of cytokines in the authors' opinion. It seems like IL1b inhibition alone only has a minor effect on MPN/fibrosis, in particular given the spleen weight. What might be the role of increased IL1b in MPN in the author's opinion.

Reviewer #3 (Remarks to the Author): with expertise in myeloproliferative neoplasms

This manuscript focuses on studying the inhibition of IL1b in myeloproliferative neoplasms. The authors measure IL-1b and IL-1RA in serum from MPN patients and also the expression of IL-1 receptors (IL1RA and IL-1RAcP) in cells from peripheral blood. They demonstrate that MPN patients have elevated levels IL-1b and the antagonist cytokine (IL-1RA) in the serum. Further analysis in granulocytes showed increased transcript levels for both (IL1b and IL1RA) in MPN samples compared to controls. The authors then showed that MPN-HSPCs expressed higher levels of the IL-1 receptors (IL1RA and IL-1RAcP) than normal controls and that this expression positively correlated with JAK2-V617F mutation allele burden. Next, the authors explored the function of IL1b in a mouse model driven by the JAKV617F mutation. For this, they generated IL1b^{-/-} JAKV617F mice and analyzed blood parameters in primary and transplanted mice. Whole body knockout of IL1b did not affect the disease course upon JAK activation. By contrast, transplantation of IL1b^{-/-} JAK2V617F cells into wild-type recipients lead to a mild reduction in splenomegaly, reticulin fibrosis and osteosclerosis. This effect was not observed when IL1b^{-/-} JAK2V617F cells were transplanted into IL1b^{-/-} recipient mice. Lastly, the authors explored the use of an anti-IL1b antibody alone or in combination with ruxolitinib. Mice treated with IL-1b antibody alone showed reductions in platelet counts and monocytes in blood and reduction of fibrosis and osteosclerosis in the bone marrow while dual treatment with ruxolitinib further decreased the bone marrow fibrosis and osteosclerosis with a reduction in pro-inflammatory cytokines in the bone marrow and plasma.

Overall, the authors show that IL1b and IL1RA and their receptor expression in HPSCs is increased in MPN samples and perform correlations with the JAKV617F allele burden. In mouse models, they showed that blockade of IL1b leads to reduction of fibrosis and osteosclerosis. This effect was primarily mediated through the IL1b effect on non-hematopoietic cells as the frequencies of hematopoietic stem cells and progenitors did not change. The data are convincing and the conclusion are largely justified. The main weakness is the absence of an experimental demonstrated mechanism to explain how IL1b blockade reduces fibrosis.

Major concerns:

1. The manuscript is mainly descriptive with little mechanism on how IL1b blockade reduces fibrosis MPNs. Can the authors expand on this point?
2. Was the expression of IL1R1 and IL-1RAcP also increased in HSPCs from their mouse model?
3. What is the significance of the expression of IL1R1 in HSPCs if IL1b deletion or blockage doesn't affect them?
4. MPN patients had increased levels of IL1b but higher levels of its antagonist IL1RA. However, in their mouse model IL1RA levels were not elevated in JAK2V617F mice. Is there functional role for IL1RA in the disease presentation?
5. Based on their experiments IL1b blockade mainly reduces fibrosis and osteosclerosis. Do these cells express higher levels of the IL1R that could explain the selective effect on these?
6. Ruxolitinib is fairly effective in reducing fibrosis in the mice, but does not have this activity in patients. The authors may want to mention that the reduction in fibrosis seen with the IL1 blockade may be similarly ineffective in patients.

Other comments

1. Analysis of fibrosis in Figure 2 and supplementary figure 3 was performed at 16 weeks not at 32 weeks as shown for the other mice. Is there a reason for which this was not performed at 32 weeks?
2. The legend to supplementary figure 5 doesn't match the data that are presented.

Responses to the Reviewers

We thank the reviewers for their helpful comments. To address their questions, we performed additional experiments and made changes in the manuscript (marked in blue). We also provide a separate file with all changes highlighted.

The following new data was added to the revised manuscript:

- **New Figure 5:** We performed experiments using *Nestin*-GFP reporter mice and found that loss of *IL-1 β* in *JAK2*-mutant donor cells prevented the destruction of nestin-positive MSCs, which correlated with reduced myelofibrosis
- **Supplementary Figure S3:** We found increased TGF β 1 levels in MPN patients. TGF β 1 is one of the key mediators of myelofibrosis.

Supplementary Figures S7a, S10a: We also found reduced levels of TGF β 1 in mice transplanted with *VF;IL-1 β ^{-/-}* bone marrow and in *VF* mice treated with anti-IL-1 β antibody.

- We performed a number of key control experiments requested by the reviewers:

We transplanted bone marrow cells from *WT* and *IL-1 β ^{-/-}* donors into *WT* and *IL-1 β ^{-/-}* recipients (**Figure 3 and Supplementary Figure S6**).

We also treated *WT* mice transplanted with *WT* bone marrow with anti-IL-1 β antibody (**Supplementary Figure S10**).

We confirmed that anti-IL-1 β antibody also reduced fibrosis in non transplanted *VF* mice (**Supplementary Figure S12**).

- **Revised Figure 3c and 3f:** We added data on the grade of fibrosis at 16weeks after transplantation.
- **Revised Figure 2a:** We added RNA sequencing data showing upregulation of genes from IL-1R pathway in *VF* versus *WT* mice.

We also made multiple changes in the presentation of the data in the text of the manuscript (highlighted in blue color) and in the revised Figures.

Below, we provide detailed point-by-point answers to the reviewers' comments and questions (our answers are in blue color).

Reviewer #1 (Remarks to the Author): with expertise in haematopoiesis and inflammatory cytokines

The authors show that IL-1 β is elevated in MPN patients with JAK2 mutations as well as in a JAK2 mutant MPN mouse model. Genetic knock out of IL-1 β from the hematopoietic stem and progenitor cells reduced bone marrow fibrosis as well as osteosclerosis in the MPN mice. However, this effect was not seen in the mice with IL-1 β knockout in the niche. Moreover, the authors show that pharmacological inhibition of the IL-1 β receptor in combination with Ruxolitinib reduces bone marrow fibrosis and osteosclerosis. The manuscript is very well written. Overall, further mechanistic insight and inclusion of key controls is needed.

MAJOR COMMENTS:

1) Figure 1C-D and Suppl Fig 1B: In the supplementary figure, the authors should consider including data for the number/percentage of just HSCs and HSPCs to get a baseline read if there are differences in just stem and progenitor cells between NC and MPN patients, before going into the IL1R1+ subset of the HSPCs/HSCs.

As requested by the reviewer, we have now added a graph to the revised Supplementary Figure S2b showing the overall frequencies of HSCs and HSPCs in peripheral blood from NC and MPN patients. The frequencies were increased in all MPN subsets, and highest in PMF.

2) Figure 3: The authors should include healthy WT cells as donors for their control transplants in both the competitive WT recipients and non-competitive IL-1 β ^{-/-} recipients.

We have performed transplantations of WT and IL-1 β ^{-/-} bone marrow cells into WT or IL-1 β ^{-/-} recipients. The new data has been added to the revised Figure 3a and 3d and Supplementary Figure S6a-d. Both WT and IL-1 β ^{-/-} bone marrow transplantations did not result in an abnormal phenotype in the recipient mice.

3) Figure 4D: the authors should include at minimum in the extended figures the data demonstrating the effect of IL-1 β blockade treatment on the WT donor cells and not just the MPN.

We transplanted WT donor cells into wildtype recipients and treated them with anti-IL-1 β antibody, as requested. We observed with anti-IL-1 β antibody only a minor decrease in lymphocyte numbers, but otherwise no effects on blood counts, spleen weight or bone marrow HSPCs (see new Supplementary Figure S10b-f).

4) The authors should provide some mechanistic insight as to how IL-1 β inhibition is reducing fibrosis in order to extend the manuscript

The mechanistic link between IL-1 β and myelofibrosis in our view is two-fold: First, platelets and megakaryocytes have been shown to be prime drivers in the pathogenesis of myelofibrosis (Villeval et al, Blood 1997; Ciurea et al, Blood 2007). IL-1 β has been shown to have a direct positive effect on megakaryopoiesis (Kimura et al, Blood 1990; Means et al, J Cell Physiol 1992; van den Oudenrijn et al, Br J Haematol 1999; Yang et al, Br J Haematol 2000) and to promote polyploidisation of megakaryocytes by activating NF κ B and MAPK signaling (Beaulieu et al, Arterioscler Thromb Vasc Biol 2014). TGF- β release from platelets was implicated as a key mediator of the pro-fibrotic process (Chagraoui et al, Blood 2002). To explore this hypothesis, we performed assays of TGF- β isoforms in serum of MPN patients and in bone marrow of MPN mice. All these data have been added to the revised manuscript. Our data is in-line with these studies and we also observed increased TGF β 1 serum levels in MPN patients (Figure 1) and VF mice (Supplementary Figure S10), and reduced TGF β 1 levels in bone marrow of VF mice with genetic or pharmacological inhibition of IL-1 β (Supplementary Figures S7a and S10a).

The second link to myelofibrosis is through the neuropathy inflicted by IL-1 β on the Schwann cells and sympathetic nerve fibers that innervate bone marrow and are necessary for maintaining Nestin-positive mesenchymal stromal cells (MSCs) (as reported by Arranz et al, Nature 2014). The prediction is that eliminating IL-1 β will prevent damage to the Schwann cells and sympathetic nerve fibers. We therefore performed transplantations with bone marrow from VF, VF;IL-1 β ^{-/-} or WT donors into Nestin-GFP reporter mice and tested the effects of genetic ablation of IL-1 β on the sympathetic neuropathy of the bone marrow niche (new Figure 5). Recipients of VF bone marrow showed severe damage of the sympathetic innervation and loss of Nestin-GFP positive cells accompanied by elevated thrombocytosis and higher grades of reticulin fibrosis, whereas recipients of VF;IL-1 β ^{-/-} bone marrow were protected from this damage and showed reduced platelet numbers as well as significant reduction in the grade of reticulin fibrosis in bone marrow (new Figure 5). A link between increase of Nestin+ MSCs and reduced myelofibrosis was also found in our clinical phase II trial using Mirabegron, a β -3 sympathomimetic agonist, that corrected the damage inflicted by the MPN clone on the Nestin+ MSCs (Drexler et al, Haematologica 2019).

5) In the text the authors claim (related to Figure 2), "Taken together, these results demonstrate that IL-1 β deficiency in this MPN mouse model reduced

inflammation in the bone marrow, but apart from slightly increasing platelet and leukocyte numbers while decreasing red cell parameters, did not affect the overall course of MPN disease.” However, these data argue against the hypothesis because there are a similar number of cells in the BM and Spleen although lowered inflammation. Is inflammation associative or causative in MPN progression?

We agree with the reviewer that the wording of this sentence was not optimal. We now changed this sentence at the end of the first paragraph on page 7 to: "Taken together, these results show that complete loss of IL-1 β in this MPN mouse model reduced inflammatory cytokines in the bone marrow, but did not affect the overall course of MPN disease."

We also agree that there are some discrepancies between the phenotypes and in the levels of pro-inflammatory cytokines in the data that we obtained. It is at present unclear why the complete loss of *IL-1 β* in non-transplanted *VF;IL-1 β ^{-/-}* knockout mice did not show a decrease in myelofibrosis compared to *VF*, despite reduction in inflammatory cytokines due to absence of IL-1 β (Figure 2d). In general, our data shows a good correlation between the platelet counts and the grade of fibrosis (Figures 3,4 and 5) and we think that the reason why non-transplanted *VF;IL-1 β ^{-/-}* mice showed no significant changes in myelofibrosis compared to *VF* is because they also showed no reduction in platelet counts (revised Figure 2b). As mentioned above, both inflammation and abnormal megakaryopoiesis are important drivers of myelofibrosis. The first (but not the latter) was reduced in non-transplanted *VF;IL-1 β ^{-/-}* mice, possibly due to developmental compensation restoring megakaryopoiesis when IL-1 β is constitutively abrogated throughout life. Although we have no definitive explanation as to why non-transplanted *VF;IL-1 β ^{-/-}* mice did not show the expected reduction in platelet counts, this might explain the unchanged myelofibrosis in nontransplanted *VF;IL-1 β ^{-/-}* mice.

IL-1 β ^{-/-} recipients transplanted with *VF;IL-1 β ^{-/-}* bone marrow also displayed a complete loss of *IL-1 β ^{-/-}*, but as expected showed lower platelet counts during the first 20 weeks after transplantation (revised Figure 3d). At 16 weeks, there was a trend towards lower grade of MF (data added to revised Figure 3f and also shown here as "Figure 1 for reviewers only"). However, after 24 weeks the platelets increased in these *VF;IL-1 β ^{-/-}* transplanted mice to levels higher than in the *VF*transplanted group and at 32 weeks there was no difference in the grade of myelofibrosis between the two groups, similar to the non-transplanted *VF;IL-1 β ^{-/-}* mice.

Figure 1 for reviewers only also revised Figure 3d and f,)(. see

Transplantation of bone marrow from
 VF or $VF;IL-1\beta^{-/-}$ donor mice into $IL-$

1β recipient mice. The time course

of platelet counts and the grade of reticulin fibrosis are shown.

WT recipients transplanted with $VF;IL-1\beta^{-/-}$ bone marrow showed lower platelet counts than WT recipients transplanted with VF bone marrow and also showed reduced grade of myelofibrosis (revised Figure 3c)

Finally, WT recipients transplanted with VF bone marrow that were treated with anti-IL-1 β antibody alone showed reduced platelet counts and also displayed reduced grade of myelofibrosis compared with vehicle (Figure 4). Overall, the correlation between the platelet counts and the grade of myelofibrosis remains good throughout our whole dataset.

6) “While loss of IL-1 β restricted to hematopoietic cells increased the levels of IL-1Ra in BM, the complete loss of IL-1 β in all tissues was not accompanied by an increase in IL-1Ra (Supplementary Figure S5), suggesting that the overall activity of IL-1 signaling is reduced when IL-1 β is lost in hematopoietic cells only.” While the increase may be trending, it is not significant. These data seem to detract from the main point of the manuscript and the authors write in the discussion that there isn’t a good explanation for them. It may be best to exclude the data from the manuscript to retain focus.

We agree with the reviewer that the IL-1Ra does not provide a convincing explanation for the observed differences in phenotypes between loss of IL-1 β in hematopoietic cells only versus in all tissues. We have therefore deleted this sentence in the revised manuscript.

MINOR COMMENTS:

7) Text edit (italicized). Overall, these results show a positive correlation between JAK2-V617F and increased IL-1 signaling in MPN patients.

In this sentence we did not italicize IL- β , because here we mean the IL-1 proteins (IL-1 β and I IL-1 α).

8)Text edit (italicized). IL-1 β is considered a master regulator of inflammation that controls the production of multiple pro-inflammatory cytokines and induces its own expression via a positive feedback loop in an autocrine or paracrine manner.

In this sentence we did not italicize IL-1 β , because here we mean the IL-1 β protein.

Reviewer #2 (Remarks to the Author): with expertise in myeloproliferative neoplasms

Rai, Skoda et al. dissect the role of IL1 β in patient samples and murine models on JAK2VF-driven MPN associated with bone marrow fibrosis. IL1b as a key immunoregulatory and proinflammatory cytokine, is produced by the inflammasome and has gained significant attention in the pathogenesis of the myeloid neoplasms in the last couple of years. The authors convincingly show that IL1 β is upregulated in human disease and also show an association with JAK2 mutated disease (in their own data set and using publicly available data sets). In particular the data on patient samples is strong and detailed and really raises the question how knockout of IL1 β affects the MPN phenotype and the development of bone marrow fibrosis. Given the title of the article, the data presented in the manuscript were rather surprising as one would have expected an amelioration of the MPN phenotype. However, the results using the genetic knockout in HSCs and stromal cells (recipient mice after transplant) show rather subtle effects. In my opinion this is still very interesting but it should be highlighted that the effect is rather minor and most effect is seen in the knockout in HSCs after transplant (maybe after addition of additional stress).

We agree with the reviewer that it is surprising and unclear why the complete loss of IL-1 β in non-transplanted VF;IL-1 β ^{-/-} knockout mice did not show the expected reduction in myelofibrosis, which was observed in WT recipients that were transplanted with VF;IL-1 β ^{-/-} knockout bone marrow. As explained in response to the similar comment by Reviewer 1 (point 4), both inflammation and abnormal megakaryopoiesis are important drivers of myelofibrosis. The first (but not the latter) was reduced in non-transplanted VF;IL-1 β ^{-/-} mice, possibly due to developmental compensation restoring megakaryopoiesis when IL-1 β is abrogated throughout life. Although we have no definitive explanation as to why non-transplanted VF;IL-1 β ^{-/-} mice did not show the expected reduction in platelet counts, this might explain the unchanged myelofibrosis in non-transplanted VF;IL-1 β ^{-/-} mice.

In general, our data shows a good correlation between the platelet counts and the grade of fibrosis (Figures 3,4 and 5). Contrary to expectations, the platelet counts in non-transplanted *VF;IL-1 β ^{-/-}* knockout mice were essentially unchanged compared to *VF*. We think that this is also the reason for the absence of reduction in myelofibrosis (Figure 2b). Nevertheless, we also observed reduction in fibrosis in non-transplanted *VF* mice treated with anti-IL-1 β antibody (Supplementary Figure S12), suggesting that additional stress after transplantation is not required to obtain this beneficial effect.

The results of the anti-IL-1 β inhibitor are interesting but also here it seems like the major effect is caused by Ruxolitinib and the combinatorial effect is only minor. More mechanistic insights on the link between JAK-STAT signaling, the inflammasome and IL-1 β will help to understand the findings. Recent work in solid tumors suggested a link between IL-1, JAK-STAT and TGF β - it would be highly interesting to analyse this mechanistic link in the context of experiments in this paper.

In the original Figure 4f statistical significance was only calculated between vehicle and the treatment groups, but not between the treatment groups. In the revised Figure 4f we now compared and plotted significant differences between the treatment groups. Anti-IL-1 β antibody alone had slightly better effect on osteosclerosis and was comparable to Ruxolitinib alone on reticulin fibrosis (revised Figure 4f). However, the combination of anti-IL-1 β antibody plus Ruxolitinib was significantly better than either drug alone (revised Figure 4f). In addition, Ruxolitinib reduced pro-inflammatory cytokines, but the combo showed an additive effect (Figure 4G).

More specifically, I have the following concerns/comments:

1) The patient data in figure 1 are convincing. Figure 1f using the publicly available data sets indicates an interesting link between TGF β and IL-1, at least they seem to change in the same direction. Can you follow up on this potential correlation?

We measured the levels of TGF- β 1, TGF- β 2 and TGF- β 3 in serum samples of MPN patients and NC (Supplementary Figure S3). We found that mean levels of serum TGF- β 1 were elevated, TGF- β 2 were unchanged and TGF- β 3 were reduced in MPN patients compared to NC (Supplementary Figure S3a). The range of the values between individual MPN patients was rather large and overall TGF- β 1 levels in serum were several logs higher than those of TGF- β 2 and TGF- β 3. TGF- β 1 and TGF- β 2 showed weak negative correlation with JAK2-V617F allele burden (Supplementary Figure S3a), but no correlation between TGF- β 1 and IL-1 β serum levels was noted (revised Supplementary Figure S3b).

Figure 2 (for reviewers only) taken from the revised Supplementary Figure S3a.

2) In general, all headlines in the manuscript summarize the main finding of the paragraph. However, on page 6, the headline rather states the approach but not the finding. This is most likely due to the fact that there is no effect of the genetic KO of IL1b on the phenotype. I was really surprised to see the results and also had to look at the figure multiple time as the statements are pretty weak instead of summarizing this as a negative result, which is also an interesting finding (but unexpected based on the title, abstract etc).

We have modified the headline on page 6, as suggested.

3) Figure 2 is also far too dense and should be focused on the main findings. 2a is composed of 13 (!) graphs. This figure should be reduced to summarize the main message.

We have reduced the number of graphs in revised Figure 2b and we have moved some of the data to the revised Supplementary Figure S4.

4) The body weights can go into the supplements and the authors should consider showing spleen/body weight.

We have plotted the spleen weights as a function of the body weight, but this did not substantially change the representation of the data (Figure 3, for reviewers only). We therefore kept the plot with the spleen weights in mg in the main manuscript, also to remain consistent with the presentations of spleen weight data in all other Figures.

- 5) Figure 2c is also composed of too many graphs. It is interesting that although IL1b is reduced in the KO, there is basically no effect on the phenotype.

We have moved the plot showing IL-1RA to supplementary Figure S4c.

- 6) The legend for 2d is missing - please also only focus on significant results in this graph for better readability

The legend to Figure 2d was added. The bone marrow of VF;IL-1β-/- mice showed significant reduction of several cytokines compared to VF mice. Although plasma cytokine levels did not differ between the two genotypes, it's good to compare the levels side by side with bone marrow and therefore we decided to keep the graph for plasma.

- 7) It would have been good to have a wild-type control in the experiment in 2a to have the baseline hematopoietic phenotype after transplant.

We assume that the reviewer meant experiment 3a. Thank you for suggesting this improvement, we have added the wildtype controls to the revised Figure 3a.

- 8) Please show also representative images of endosteal areas for the IL1b knockout in 3c - this is the area that's affected first by fibrosis and is not shown here (not representative, comparable).

We have taken new images of the endosteal areas of the bone marrow. The representative images from 2 mice are shown in a new Supplementary Figure S8.

- 9) It is confusing that the grade of reticulin fibrosis in VF is 2, but 92% of mice had osteosclerosis? How was this calculated?

Grading of fibrosis was based on the main (average) grade present (in analogy to diagnostic assessment of human bone marrow biopsies). The mean grade of mice analyzed per group is shown. In contrast, osteosclerosis was scored as either being present or absent. Thus 92% means that 11/12 mice showed osteosclerosis in the VF group, while 38% means that 3/8 mice showed osteosclerosis in the VF;IL-1 β -/group. Although most mice had an average grade of 2, many mice with osteosclerosis demonstrated foci of fibrosis grade 3. This is why the fibrosis grade appears to be in conflict with the presence of osteosclerosis.

- 10) It is interesting that there is no stromal/recipient KO of IL1b emphasizing that the effect is mainly on HSCs as also the patient samples in 1 would have suggested. Did the authors ever check for a correlation of IL1b to the fibrosis grade (in all their (patient?) samples independent of the MPN entity).

This is a very good suggestion. Unfortunately, only a minority of samples analyzed for IL-1 β serum levels in Figure 1 were from the time of diagnosis and thus we cannot examine this potential correlation in our dataset.

- 11) The representation of fibrosis grades in 4b is suboptimal, please improve.

We have changed the presentation of the myelofibrosis grades in the revised Figure 4b to the same style as Figure 4f.

- 12) The most striking finding in the blood counts after inhibitor treatment seemed to be on monocytes. Did the authors further look into this?

This is a good suggestion. We tried to perform immuno-staining monocyte-derived fibrocytes (collagen-I and CD11b) in paraffin sections of bone marrow. However, we were unable to obtain good quality staining in our sections and were unable to enumerate these cells. IL-1 β was shown to directly accelerate myeloid differentiation via activation of a PU.1-dependent gene program (Pietras et al, Nat. Cell Biol. 2016)

- 13) It is confusing that the inhibitor does not have an effect of spleen weight but the reticulin grade is reduced. How do the authors explain this?

Spleen weight mostly correlated with the expansion of erythropoiesis, which was not strongly affected by anti-IL-1 β antibody.

14) The megakaryocytes in the anti IL1b treatment still look very abnormal. Who quantified the fibrosis grade? Was this blinded? Multiple reviewers?

All BM biopsies were analyzed by an experienced hemato-pathologist (Stefan Dirnhofer) who was completely blinded for the different treatment groups, while analyzing osteosclerosis and assigning the grade of fibrosis.

15) Were the reticulin images in 4f taken in comparable locations of the bone?

Yes, the reticulin images in 4f were taken in comparable locations.

16) In the quantification of cytokines, the major effect seems to be caused by Ruxolitinib not IL1b inhibition. What is the cellular source of cytokines in the authors' opinion. It seems like IL1b inhibition alone only has a minor effect on MPN/fibrosis, in particular given the spleen weight. What might be the role of increased IL1b in MPN in the author's opinion.

Anti-IL-1 β antibody alone or in combo was slightly better than ruxolitinib alone in reducing levels of IL-1 β , IL-6 and IL-5 in BM and plasma (Figure 4g). It has been previously shown (Kleppe et al, Cancer Discovery 2015) that both mutant (cell autonomous) and non-mutant cells can produce inflammatory cytokines and together contribute to the development of myelofibrosis. Our data indicate that the source of elevated IL-1 β production were *JAK2*-mutant cells. However, we have not measured production of IL-1 β in purified cell populations. Anti-IL-1 β antibody had comparable effect on reticulin fibrosis and the effect was slightly better on osteosclerosis (Figure 4f).

Reviewer #3 (Remarks to the Author): with expertise in myeloproliferative neoplasms

This manuscript focuses on studying the inhibition of IL1b in myeloproliferative neoplasms. The authors measure IL-1b and IL-1RA in serum from MPN patients and also the expression of IL-1 receptors (IL1RA and IL-1RAcP) in cells from peripheral blood. They demonstrate that MPN patients have elevated levels IL-1b and the antagonist cytokine (IL-1RA) in the serum. Further analysis in granulocytes showed increased transcript levels for both (IL1b and IL1RA) in MPN samples compared to controls. The authors then showed that MPN-HSPCs expressed higher levels of the IL-1 receptors (IL1RA and IL-1RAcP) than normal controls and that this expression positively correlated with JAK2-V617F mutation allele burden. Next, the authors explored the function of IL1b in a mouse model driven by the JAKV617F mutation. For this, they generated IL1b^{-/-} JAKV617F mice and analyzed blood parameters in primary and transplanted mice. Whole body knockout of IL1b did not affect the disease course upon JAK activation. By contrast, transplantation of IL1b^{-/-} JAK2V617F cells into wild-type recipients lead to a mild reduction in splenomegaly, reticulin fibrosis and osteosclerosis. This effect was not observed when IL1b^{-/-} JAK2V617F cells were transplanted into IL1b^{-/-} recipient mice. Lastly, the authors explored the use of an anti-IL1b antibody alone or in combination with ruxolitinib. Mice treated with IL-1b antibody alone showed reductions in platelet counts and monocytes in blood and reduction of fibrosis and osteosclerosis in the bone marrow while dual treatment with ruxolitinib further decreased the bone marrow fibrosis and osteosclerosis with a reduction in pro-inflammatory cytokines in the bone marrow and plasma.

Overall, the authors show that IL1b and IL1RA and their receptor expression in HPSCs is increased in MPN samples and perform correlations with the JAKV617F allele burden. In mouse models, they showed that blockade of IL1b leads to reduction of fibrosis and osteosclerosis. This effect was primarily mediated through the IL1b effect on non-hematopoietic cells as the frequencies of hematopoietic stem cells and progenitors did not change. The data are convincing and the conclusion are largely justified. The main weakness is the absence of an experimental demonstrated mechanism to explain how IL1b blockade reduces fibrosis.

Major concerns:

- 1) The manuscript is mainly descriptive with little mechanism on how IL1b blockade reduces fibrosis MPNs. Can the authors expand on this point?

The mechanistic link between IL-1 β and myelofibrosis in our view is two-fold: First, platelets and megakaryocytes have been shown to be prime drivers in the pathogenesis of myelofibrosis (Villevall et al, Blood 1997; Ciurea et al, Blood 2007).

IL-1 β has been shown to have a direct positive effect on megakaryopoiesis (Kimura et al, Blood 1990; Means et al, J Cell Physiol 1992; van den Oudenrijn et al, Br J Haematol 1999; Yang et al, Br J Haematol 2000) and to promote polyploidisation of megakaryocytes by activating NF κ B and MAPK signaling (Beaulieu et al, Arterioscler Thromb Vasc Biol 2014). TGF- β release from platelets was implicated as a key mediator of the pro-fibrotic process (Chagraoui et al, Blood 2002). To explore this hypothesis, we performed assays of TGF- β isoforms in serum of MPN patients and in bone marrow of MPN mice. All these data have been added to the revised manuscript. Our data is in-line with these studies and we also observed increased TGF β 1 serum levels in MPN patients (Figure 1) and VF mice (Supplementary Figure S10), and reduced TGF β 1 levels in bone marrow of VF mice with genetic or pharmacological inhibition of IL-1 β (Supplementary Figures S7a and S10a).

The second link to myelofibrosis is through the neuropathy inflicted by IL-1 β on the Schwann cells and sympathetic nerve fibers that innervate bone marrow and are necessary for maintaining Nestin-positive mesenchymal stromal cells (MSCs) (as reported by Arranz et al, Nature 2014). The prediction is that eliminating IL-1 β will prevent damage to the Schwann cells and sympathetic nerve fibers. We therefore performed transplantations with bone marrow from VF, VF;IL-1 β ^{-/-} or WT donors into Nestin-GFP reporter mice and tested the effects of genetic ablation of IL-1 β on the sympathetic neuropathy of the bone marrow niche (new Figure 5). Recipients of VF bone marrow showed severe damage of the sympathetic innervation and loss of Nestin-GFP positive cells accompanied by elevated thrombocytosis and higher grades of reticulin fibrosis, whereas recipients of VF;IL-1 β ^{-/-} bone marrow were protected from this damage and showed reduced platelet numbers as well as significant reduction in the grade of reticulin fibrosis in bone marrow (new Figure 5). A link between increase of Nestin⁺ MSCs and reduced myelofibrosis was also found in our clinical phase II trial using Mirabegron, a β -3 sympathomimetic agonist, that corrected the damage inflicted by the MPN clone on the Nestin⁺ MSCs (Drexler et al, Haematologica 2019).

2) Was the expression of IL1R1 and IL-1RAcP also increased in HSPCs from their mouse model?

We tried to use antibodies against mouse IL1R1 and IL-1RAcP for flow cytometry, but unfortunately, they did not work as well as the anti-human antibodies and we were unable to determine the expression of these proteins on the cell surface of mouse HSPCs. However, RNAseq analysis showed increased expression of IL-1 signaling pathway genes, including IL1R1 and IL-1RAcP, in HSPCs from VF mice compared to wildtype controls. We included this new data in revised Figure 2a.

3) What is the significance of the expression of IL1R1 in HSPCs, if IL1b deletion or blockage doesn't affect them?

The expression of IL1R1 on HSPCs is not only required for IL-1 β signaling, but also transmits signaling by IL-1 α and therefore expression levels could be maintained for that reason. The frequencies of pre-MegE were reduced in bone marrow and spleen of VF mice deficient for IL-1 β (Supplementary Figure S7b), in line with the reported positive effects of IL-1 β on megakaryopoiesis and platelet production (Kimura et al., 1990; van den Oudenrijn et al., 1999; Means et al., 1992).

4) MPN patients had increased levels of IL1b but higher levels of its antagonist IL1RA. However, in their mouse model IL1RA levels were not elevated in JAK2V617F mice. Is there functional role for IL1RA in the disease presentation?

At present, we do not have an explanation for the difference in IL-1RA expression between humans and mice. In humans, IL-1RA was elevated mainly in PV and PMF and less so in ET (revised Figure 1b). IL-1RA expression data in mice were determined at 16 weeks post induction, when VF mice are only beginning to display myelofibrosis (previously Figure 2c, now revised Supplementary Figure S4d). We do not have data from later time points.

5) Based on their experiments IL1b blockade mainly reduces fibrosis and osteosclerosis. Do these cells express higher levels of the IL1R that could explain the selective effect on these?

We have not determined the expression of IL-1R in bone marrow on pro-fibrotic fibroblasts or cells of the osteoblast lineage in VF mice. However, it has been previously shown in a CML model that cells of the osteoblast lineage have high expression of IL-1R1 (Schepers et al, Cell Stem Cell 2013).

As stated in our response to point 1 above, our current model is that the effects of IL1 β on myelofibrosis are primarily driven by IL-1 β increasing pro-fibrotic cells of the megakaryocytic lineage and the neuropathy inflicted by IL-1 β resulting in destruction of nestin-positive MSCs.

6) Ruxolitinib is fairly effective in reducing fibrosis in the mice, but does not have this activity in patients. The authors may want to mention that the reduction in fibrosis seen with the IL1 blockade may be similarly ineffective in patients.

The effect of ruxolitinib on fibrosis in preclinical mouse models is variable and appears to mainly dependent on the dose and duration of ruxolitinib treatment.

Higher doses of ruxolitinib (60-90 mg/kg, BID) or longer treatment regimen (more than 3 weeks) were shown to be effective in reducing myelofibrosis in mice (Kleppe et al. 2018, Cancer Cell; Brkic et al., 2021 Leukemia). In contrast, shorter treatment regimen and/or lower dose (30 mg/kg; 3 weeks, QD) of ruxolitinib did not reduce myelofibrosis (Wen et al., 2015 Nature Medicine). We used ruxolitinib at 30 mg/kg BID, but we treated for 8 weeks. Thus, our results are comparable with the other reports.

The data presented in Figure 4 were obtained in mice transplanted with bone marrow from *VF* mice. However, we also tested the effects of anti-IL-1 β antibody in non-transplanted *VF* and *WT* mice (new Supplementary Figure S12). Treatment with anti-IL-1 β antibody started 28-weeks after tamoxifen induction and significantly reduced the grade of reticulin fibrosis in *VF* mice compared to isotype controls.

We agree that results obtained in our *VF* mouse model do not guarantee that treatment with anti-IL-1 β antibody will also be effective in MPN patients. Nevertheless, our preclinical data provide a rationale to explore these effects further and test anti-IL-1 β antibodies in a clinical trial.

Other comments

7) Analysis of fibrosis in Figure 2 and supplementary figure 3 was performed at 16 weeks not at 32 weeks as shown for the other mice. Is there a reason for which this was not performed at 32 weeks?

We have now analyzed non-transplanted mice at 32 weeks after tamoxifen induction and added this data in revised Figure 2b and Supplementary Figure S5a. Although the average grade of reticulin fibrosis was higher at 32 weeks compared to 16 weeks post tamoxifen induction, there was still no difference between the *VF* and *VF*;IL-1 β mice.

8) The legend to supplementary figure 5 doesn't match the data that are presented.

We have corrected the legend to original Supplementary Figure 5 (now revised Supplementary Figure S7).

REVIEWERS' COMMENTS

Reviewer #1 (Remarks to the Author):

The authors have adequately improved their manuscript based on my prior comments and have satisfied my concerns.

Reviewer #2 (Remarks to the Author):

In the revised version of the manuscript, and also the reply to my comments, the authors have addressed all my concerns sufficiently. I do not have further questions.

Reviewer #3 (Remarks to the Author):

The authors have addressed my concerns.

RESPONSES TO THE

REVIEWERS' COMMENTS

Reviewer #1 (Remarks to the Author):

The authors have adequately improved their manuscript based on my prior comments and have satisfied my concerns.

Thank you very much for raising important points and for helping to improve our manuscript.

Reviewer #2 (Remarks to the Author):

In the revised version of the manuscript, and also the reply to my comments, the authors have addressed all my concerns sufficiently. I do not have further questions.

Thank you very much for raising important points and for helping to improve our manuscript.

Reviewer #3 (Remarks to the Author):

The authors have addressed my concerns.

Thank you very much for raising important points and for helping to improve our manuscript.